# Distributional Training Data Attribution: What do Influence Functions Sample?

**Bruno Mlodozeniec**[♦][1][2]   **Isaac Reid**[♦][1]   **Sam Power**[3]   **David S. Krueger**[4]
**Murat A. Erdogdu**[♣][5][6]   **Richard E. Turner**[♦][1][7]   **Roger B. Grosse**[♣][5][6]

[1]University of Cambridge   [2]Max Planck Institute for Intelligent Systems
[3]University of Bristol   [4]Mila - Quebec AI Institute   [5]University of Toronto
[6]Vector Institute   [7]Alan Turing Institute

bkm28@cam.ac.uk   ir337@cam.ac.uk

## Abstract

Randomness is an unavoidable part of training deep learning models, yet something that traditional training data attribution algorithms fail to rigorously account for. They ignore the fact that, due to stochasticity in the initialisation and batching, training on the same dataset can yield different models. In this paper, we address this shortcoming through introducing *distributional* training data attribution (d-TDA), the goal of which is to predict how the distribution of model outputs (over training runs) depends upon the dataset. Intriguingly, we find that *influence functions* (IFs), a popular data attribution tool, are 'secretly distributional': they emerge from our framework as the limit to unrolled differentiation, without requiring restrictive convexity assumptions. This provides a new perspective on the effectiveness of IFs in deep learning. We demonstrate the practical utility of d-TDA in experiments, including improving data pruning for vision transformers and identifying influential examples with diffusion models.

## 1 Introduction

Training data attribution (TDA) techniques are of fundamental interest in machine learning, shedding light on the relationship between a model's properties and its training data. TDA is typically framed as a counterfactual prediction problem: estimating how a model's behaviour would change upon removal of particular examples from the training dataset [1, 2] . This invites the concept of *influence*. Training examples are deemed 'influential' if the model's behaviour would change significantly upon their exclusion. The practical utility of TDA has been demonstrated in applications including interpreting, debugging and improving models [2, 3], dataset curation [4], and data valuation [1, 5].

**Influence Functions**. It is typically prohibitively expensive to compute influence by retraining with different datapoints removed. This has motivated a number of TDA methods designed to approximate influence, but without actually retraining. Amongst such TDA methods, a leading example is *influence functions* (IFs) [2, 6]. This classical technique from robust statistics uses the implicit function theorem to estimate the optimal model parameters' sensitivity to downweighting a training datapoint. IFs have been deployed to investigate the generalisation patterns of 52 billion parameter large language models [3], and for data attribution of diffusion models [7]. Separately, researchers have proposed an alternative TDA method called *unrolled differentiation* [8, 9, 10]. Here, one differ-

---

♦ Equal contribution first authors. Order decided by who can swim the furthest underwater.   ♣ Shared senior authors.

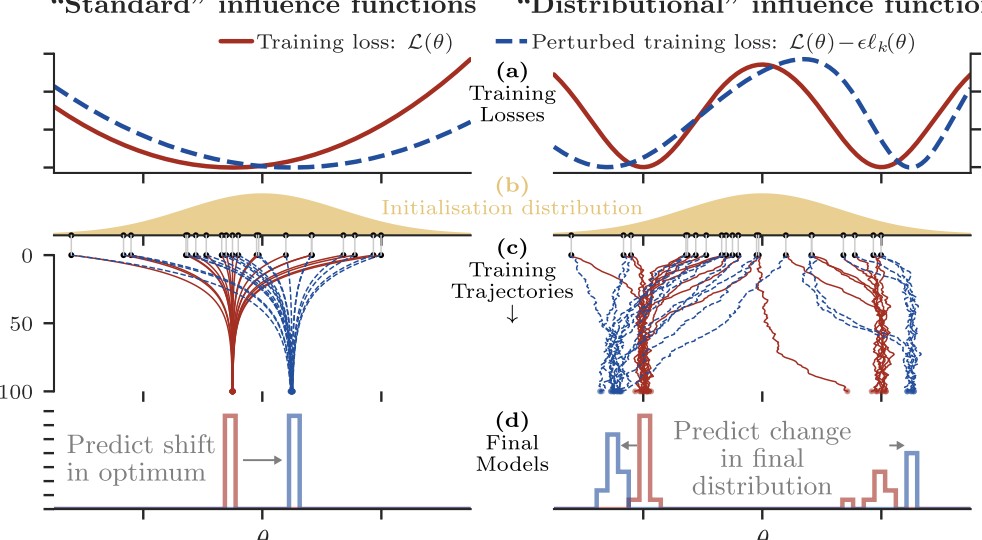

**“Standard” influence functions**     **“Distributional” influence functions**

— Training loss: $\mathcal{L}(\theta)$     - - Perturbed training loss: $\mathcal{L}(\theta) - \epsilon \ell_k(\theta)$

**(a)** Training Losses

**(b)** Initialisation distribution

**(c)** Training Trajectories ↓

Predict shift in optimum

**(d)** Final Models

Predict change in final distribution

$\theta$         $\theta$

Figure 1: **Distributional training data attribution**. Classical data attribution methods like influence functions are typically motivated using convex loss functions, predicting a deterministic shift to the unique optimal model weights *(left)*. In contrast, this paper advocates for a *distributional* perspective, approximating the new probability distribution over model parameters/outputs after removal of training examples (interpreted as perturbation of the training loss). This includes for non-convex loss functions *(right)*.

entiates through a particular training trajectory to directly obtain the sensitivity of the final model parameters to the weighting of a particular example in the loss function. Unrolled differentiation tends to work better than IFs in experiments, but it is more expensive to compute.

**Randomness in training**. The success of IFs in deep learning is perhaps surprising because the classical foundations of both TDA and IFs fail to account for a core property of modern training: stochasticity [11]. Given the randomness inherent in weight initialisation and mini-batching, training can be understood as *sampling* from a distribution over final models. Each training run corresponds to drawing a single sample from this distribution. Yet classical TDA is only defined for deterministic training algorithms, and IFs are primarily understood for convex objectives (or by finding convex proxies [11]). Stochasticity is usually dismissed as a nuisance for TDA methods, glossed over in method derivations [2]. At best, it is sometimes heuristically managed by ensembling or averaging [12]. In practice, stochasticity makes it difficult to diagnose which changes to model behaviour are attributable to changes in the training dataset, and which are due to sampling randomness.

**Introducing *distributional* training data attribution**. In this paper, we argue that the randomness in model training is not a nuisance. Conversely, it ought to play a central role in our understanding of influence, and deserves a proper mathematical treatment. Viewing training as sampling from a distribution over final model weights (or outputs), the goal of TDA should be to efficiently predict changes to this distribution under modifications to the training dataset: a novel perspective that we coin *distributional* training data attribution (d-TDA). Figure 1 provides a visual schematic.

**Influence functions are distributional**. We show that unrolled differentiation is natively a d-TDA method (Section 3.1). Subsequently, in Section 4.1, we rigorously show that IFs *approximate* unrolled differentiation for long enough training times, and hence *IFs are already inherently distributional*.

**Core contributions**. (1) We introduce *distributional training data attribution* (d-TDA), a framework for studying data attribution in stochastic deep learning settings (Section 3). (2) We show that influence functions (IFs) are 'secretly distributional', solving special limiting cases of a d-TDA task (Section 4). This may help explain the effectiveness of IFs in deep learning, far from the convex setting in which they were originally proposed. (3) We propose *distributional influence*, which quantifies the importance of examples by how much their inclusion/exclusion affects the distribution over model weights and outputs. We show that distributional influence captures interesting information missing from its regular predecessor, and leads to more effective data pruning (Section 5).

## 2 Background

**'Classical' Training Data Attribution (TDA).** Consider the space $\mathfrak{D} := \cup_{N=1}^{\infty} \mathcal{Z}^N$ of possible finite training datasets $\mathcal{D} := (z_i)_{i=1}^N$. In classical TDA, one is concerned with *deterministic* training algorithms $\boldsymbol{\theta}^* : \mathfrak{D} \to \mathbb{R}^{d_{\text{param}}}$, which take a dataset as their input and return 'trained' model parameters $\boldsymbol{\theta}^*(\mathcal{D})$. The goal of TDA is to predict how the output of the training algorithm $\boldsymbol{\theta}^*$ would change if it were run using a perturbed training dataset $\mathcal{D}'$, with some examples removed. Concretely, given some trained model $\boldsymbol{\theta}^*(\mathcal{D})$, TDA methods $\tilde{\boldsymbol{\theta}}^*(\mathcal{D}')$ aim to approximate $\boldsymbol{\theta}^*(\mathcal{D}') \approx \tilde{\boldsymbol{\theta}}^*(\mathcal{D}')$ *without* actually retraining the model. Of course, in practice, one is typically interested in the change in some *measurement function* $m : \mathbb{R}^{d_{\text{param}}} \to \mathbb{R}^{d_{\text{m}}}$ when the dataset is modified — for instance, the loss on a particular test example. Therefore, TDA methods $\tilde{\boldsymbol{\theta}}^*(\cdot)$ are typically evaluated on their ability to approximate $m(\boldsymbol{\theta}^*(\mathcal{D}')) \approx m\big(\tilde{\boldsymbol{\theta}}^*(\mathcal{D}')\big)$.

**'Classical' influence.** The discussion above invites the concept of *influence*. The influence of an example is the change in the measurement $m \circ \boldsymbol{\theta}^*$ when the example is removed from the training dataset. Influential samples change the measurement by a large amount. The influence of a training datapoint $z_k$ with respect to a measurement function $m$ is given by:

$$\mathtt{Inf}(z_k) := m(\boldsymbol{\theta}^*(\mathcal{D})) - m(\boldsymbol{\theta}^*(\mathcal{D} \setminus z_k)). \tag{1}$$

This is extended to groups of examples $(z_i)_{k=1}^{N_k} \subset \mathcal{D}$ in the obvious way. To approximate $\mathtt{Inf}(z_k)$ without actually retraining, one uses a TDA method to approximate $\boldsymbol{\theta}^*(\mathcal{D} \setminus z_k)$.

**Response.** A practical difficulty posed by the formulation of influence in Eq. (1) is that $\mathfrak{D}$, the domain of the training algorithm $\boldsymbol{\theta}^*(\cdot)$, is discontinuous. Datapoints $z_k$ are either included or not included. The binary nature of this choice makes it difficult to analyse $\mathtt{Inf}(z_k)$ directly using gradient-based methods. Hence, it is typical to instead consider a *continuous relaxation to the training algorithm*.

Let us introduce a scalar $\varepsilon \in \left[0, \frac{1}{N}\right]$ which controls the *weighting* of a particular example in the training algorithm. Suppose $\varepsilon = 0$ corresponds to inclusion and $\varepsilon = \frac{1}{N}$ corresponds to exclusion, with intermediate values meaning the example is still present but downweighted. The precise setup will depend on the training algorithm of interest. Let $\boldsymbol{\theta}^*_{\mathcal{D} \to \mathcal{D} \setminus z_k}(\varepsilon)$ denote the (assumed deterministic) outcome of the training algorithm with loss $\mathcal{L}_{\mathcal{D} \to \mathcal{D} \setminus z_k}(\varepsilon)$. Provided $\boldsymbol{\theta}^*_{\mathcal{D} \to \mathcal{D} \setminus z_k}(\varepsilon)$ is continuous and twice-differentiable at $\varepsilon = 0$, we have that

$$\boldsymbol{\theta}^*_{\mathcal{D} \to \mathcal{D} \setminus z_k}(\varepsilon) = \boldsymbol{\theta}^*(\mathcal{D}) + \varepsilon \underbrace{\frac{\mathrm{d}\boldsymbol{\theta}^*_{\mathcal{D} \to \mathcal{D} \setminus z_k}(\varepsilon)}{\mathrm{d}\varepsilon}\bigg|_{\varepsilon=0}}_{\text{Response } \boldsymbol{r}(z_k) :=} + O(\varepsilon^2) \quad \text{as } \varepsilon \to 0. \tag{2}$$

Hence, we define the *response* $\boldsymbol{r}(z_k) := \frac{\mathrm{d}\boldsymbol{\theta}^*_{\mathcal{D} \to \mathcal{D} \setminus z_k}(\varepsilon)}{\mathrm{d}\varepsilon}|_{\varepsilon=0}$, such that $m(\boldsymbol{\theta}^*(\mathcal{D} \setminus z_k)) = m(\boldsymbol{\theta}^*(\mathcal{D})) + \varepsilon \nabla m^\top \boldsymbol{r}(z_k) + \mathcal{O}(\varepsilon^2)$. Intuitively, response measures the sensitivity of the training algorithm output with respect the weighting $\varepsilon$ of the example $z_k \in \mathcal{D}$. To make this more explicit, we will now give two concrete examples: *influence functions* and *unrolled differentiation*.

**1. Influence functions.** Many classical algorithms only depend on the data through a loss function $\mathcal{L}_{\mathcal{D}}(\boldsymbol{\theta}) := \frac{1}{N} \sum_{n=1}^N \ell_n(\boldsymbol{\theta})$ with $\ell_n : \mathbb{R}^{d_{\text{param}}} \to \mathbb{R}$ some per-example loss. One natural way to codify downweighting in that case is to define an *interpolated* loss $\mathcal{L}_{\mathcal{D} \to \mathcal{D} \setminus z_k}(\varepsilon) := \mathcal{L}_{\mathcal{D}} - \varepsilon \ell_k$. Suppose that the loss function $\mathcal{L}_{\mathcal{D}}$ has a single unique minimum and that the training algorithm successfully locates it. Mathematically, this can be written as $\boldsymbol{\theta}^*(\mathcal{D}) = \arg\min_{\boldsymbol{\theta} \in \mathbb{R}^d} \mathcal{L}_{\mathcal{D}}(\boldsymbol{\theta})$. Minimising the interpolated loss $\mathcal{L}_{\mathcal{D} \to \mathcal{D} \setminus z_k}(\varepsilon)$ and applying the implicit function theorem, it is straightforward to prove that in this special case:

$$\boldsymbol{r}(z_k) = \nabla^2 \mathcal{L}_{\mathcal{D}}(\boldsymbol{\theta}^*(\mathcal{D}))^{-1} \nabla \ell_k(\boldsymbol{\theta}^*(\mathcal{D})) =: \boldsymbol{r}_{\text{IF}}. \tag{3}$$

We derive this result in detail in Section B. $\boldsymbol{r}_{\text{IF}}$ is referred to as an *influence function* (IF) – a popular TDA tool. The effectiveness of IFs for deep learning is perhaps surprising given the unrealistic assumptions made during their derivation.

**2. Unrolled differentiation.** Suppose instead that the model is trained using *stochastic gradient descent* (SGD). Consider the weight update rule $\boldsymbol{\theta}_{t+1} = \boldsymbol{\theta}_t - \frac{1}{B} \sum_{n=1}^N \delta_n^t \nabla \ell_n(\boldsymbol{\theta}_t)$, where $(\boldsymbol{\theta}_t)_{t \in \mathbb{N}}$

denotes the trajectory of model parameters and $\boldsymbol{\theta}_0$ is some random initialisation. $\boldsymbol{\delta}^t$ with $t \in \mathbb{N}$ are independently and identically distributed batching variables in $\{0,1\}^N$, with mean $\mathbb{E}[\delta_n^t] = \frac{B}{N}$. In close analogy to the interpolated loss $\mathcal{L}_{\mathcal{D} \to \mathcal{D} \setminus z_k}(\varepsilon)$, consider the interpolated update step:[1]

$$\boldsymbol{\theta}_{t+1}(\varepsilon) = \boldsymbol{\theta}_t(\varepsilon) - \frac{\eta_t}{B} \sum_{n=1}^{N} \delta_n^t \nabla \ell_n(\boldsymbol{\theta}_t)(1 - \varepsilon \mathbb{1}_{n=k}), \tag{4}$$

where $\mathbb{1}_{n=k}$ is the indicator function. If we train for $T$ timesteps, one can *directly* differentiate through the training trajectory to obtain the sensitivity of the final model weights $\boldsymbol{\theta}_T$ with respect to the weighting $\varepsilon$. Applying the chain rule, one obtains a rather cumbersome expression (Eq. (57) in Section D). In this setting, we call $\boldsymbol{r}_{\text{UD}} := \frac{\mathrm{d}\boldsymbol{\theta}_T}{\mathrm{d}\varepsilon}\big|_{\varepsilon=0}$ the *unrolled differentiation* response. $\boldsymbol{r}_{\text{UD}}$ can be used as a classical TDA method if we consider all sources of randomness to be fixed. This algorithm tends to work better than IFs in experiments, but the repeated computation and caching of Hessians makes its naive implementation expensive for long training runs. This has prompted work on *approximate* unrolled differentiation [9].

## 3 Distributional Training Data Attribution

An obvious problem with the classical TDA formulation described in Section 2 is that in reality training is *stochastic*: the randomness in model initialisation and SGD precludes defining a deterministic map $\boldsymbol{\theta}^* : \mathcal{D} \to \mathbb{R}^{d_{\text{param}}}$. Even retraining with an identical dataset will in general give a different model; $\boldsymbol{\theta}^*(\mathcal{D})$ is better thought of as a random variable. Previous work has dealt with this randomness heuristically by averaging over training ensembles [2, 9]. In contrast, in this paper we advocate for a more rigorous distributional perspective. Taking the model initialisation $\boldsymbol{\theta}_0$ and the batch selections $(\boldsymbol{\delta}_t)_{t \in \mathbb{N}}$ to be random variables on some probability space $(\Omega, \mathcal{F}, \mathbb{P})$, we frame *distributional* training data attribution as follows.

> **Distributional training data attribution (d-TDA)**. Let $\boldsymbol{\theta}^*(\mathcal{D})$ be the outcome of (stochastic) training with some dataset $\mathcal{D} \in \mathfrak{D}$. Let $\mu_{\mathcal{D}}(A) := \mathbb{P}[\boldsymbol{\theta}^*(\mathcal{D}) \in A]$ (for $A \in \mathcal{F}$) denote its probability distribution. Let $m_{\#}\mu_{\mathcal{D}}$ be the distribution of some measurement function $m : \mathbb{R}^{d_{\text{param}}} \to \mathbb{R}^{d_{\text{m}}}$ of the trained model. The goal of *distributional* TDA is to reason about the behaviour of $\mu_{\mathcal{D}}$ and $m_{\#}\mu_{\mathcal{D}}$ with respect to changing $\mathcal{D}$ – especially, removing examples by taking $\mathcal{D} \to \mathcal{D} \setminus z_k$.

Rather than considering randomness to be a nuisance, d-TDA acknowledges that the training dataset determines the distribution over trained models. Effective d-TDA methods answer questions like:

1. Given samples from $\mu_{\mathcal{D}}$, how can I approximately sample from $\mu_{\mathcal{D} \setminus z_k}$?
2. If removed from the training dataset, which example $z_k \in \mathcal{D}$ would most drastically change $\mu_{\mathcal{D}}$?
3. Which examples should I remove to change the variance of $m_{\#}\mu_{\mathcal{D}}$ upon retraining?

The fact that d-TDA predicts changes in distributions over measurements leads us to reevaluate the notion of influence. In particular, removing influential samples ought to substantially modify $m_{\#}\mu_{\mathcal{D}}$. With this in mind, we define *distributional influence* (c.f. Eq. (3)) as follows:

**Definition 1. (Distributional influence)**. The distributional influence of a training example $z_k \in \mathcal{D}$ with respect to a measurement function $m : \mathbb{R}^{d_{\text{param}}} \to \mathbb{R}^{d_{\text{m}}}$ is given by:

$$\texttt{DistInf}(z_k) := \Delta\big(m_{\#}\mu_{\mathcal{D}} \| m_{\#}\mu_{\mathcal{D} \setminus z_k}\big), \tag{5}$$

where $\Delta(\mu_1 \| \mu_2)$ is some 'difference function' between $\mu_1$ and $\mu_2$.

There exist many possible instantiations of distributional influence, depending on the choice of $\Delta$. Letting $X \sim \mu_1, Y \sim \mu_2$ denote the final measurement random variables, one could consider:

$$\Delta(\mu_1 \| \mu_2) := \begin{array}{c|ccc} & \textit{Mean influence} & \textit{Variance increase influence} & \textit{Wasserstein influence} \\ & \mathbb{E}(X) - \mathbb{E}(Y) & \text{Var}(Y) - \text{Var}(X) & \mathcal{W}_2(\mu_1, \mu_2) \end{array}$$

### 3.1 Distributional influence with unrolled differentiation

To compute $\texttt{DistInf}(z_k)$, we need to (approximately) sample from $\mu_{\mathcal{D} \setminus z_k}$ without retraining the model. This can be achieved using unrolled differentiation, described by the pseudocode below.

---

[1]This can be roughly thought of as SGD updates with the interpolated loss function $\mathcal{L}_{\mathcal{D} \to \mathcal{D} \setminus z_k}(\varepsilon)$.

**Alg. 1. Unrolled differentiation for d-TDA**.

1. Sample $\boldsymbol{\theta}^*(\mathcal{D}) := \boldsymbol{\theta}_T$ from $\mu_{\mathcal{D}}$ by training the model with stochastic updates (Eq. (4)).
2. Obtain approximate samples from $\mu_{\mathcal{D} \setminus z_k}$ *without retraining* by taking $\boldsymbol{\theta}^*(\mathcal{D} \setminus z_k) \approx \boldsymbol{\theta}^*(\mathcal{D}) + \frac{1}{N}\boldsymbol{r}_{\text{UD}}$, with $\boldsymbol{r}_{\text{UD}} := \frac{\mathrm{d}\boldsymbol{\theta}_T}{\mathrm{d}\varepsilon}\big|_{\varepsilon=0}$ the unrolled differentiation response. If interested in the distribution over some measurement, compute $m(\boldsymbol{\theta}^*(\mathcal{D} \setminus z_k)) \approx m(\boldsymbol{\theta}^*(\mathcal{D})) + \frac{1}{N}\nabla m(\boldsymbol{\theta}^*(\mathcal{D}))^\top \boldsymbol{r}_{\text{UD}}$.
3. Using these two sets of (correlated) samples, compute the difference function $\Delta$ between the empirical distributions to efficiently approximate $\texttt{DistInf}(z_k)$.

When computing $\boldsymbol{r}_{\text{UD}}$, the following observation simplifies differentiating through long training trajectories.

**Remark 1. (Unrolled differentiation is a Markov chain)**. Applying the chain rule of differentiation to Eq. (4) gives the following recursive formula for $(\boldsymbol{\theta}_t, \boldsymbol{r}_t) := \left(\boldsymbol{\theta}_t, \frac{\mathrm{d}\boldsymbol{\theta}_t}{\mathrm{d}\varepsilon}\big|_{\varepsilon=0}\right)$:

$$\begin{pmatrix} \boldsymbol{\theta}_{t+1} \\ \boldsymbol{r}_{t+1} \end{pmatrix} = \begin{pmatrix} \boldsymbol{\theta}_t - \frac{\eta_t}{B}\sum_{n=1}^N \delta_n^t \nabla \ell_n(\boldsymbol{\theta}_t) \\ \left(I - \frac{\eta_t}{B}\sum_{n=1}^N \delta_n^t \nabla^2 \ell_n(\boldsymbol{\theta}_t)\right)\boldsymbol{r}_t + \frac{\eta_t}{B}\delta_k^t \nabla \ell_k(\boldsymbol{\theta}_t) \end{pmatrix}. \tag{6}$$

Intuitively, Eq. (6) shows that the response $\boldsymbol{r}_{t+1}$ depends on the response at the previous timestep $\boldsymbol{r}_t$, modulated by the loss function curvature. If datapoint $z_k$ is present in the batch sampled at timestep $t$, $\boldsymbol{r}_{t+1}$ also depends on the corresponding loss gradient $\nabla \ell_k(\boldsymbol{\theta}_t)$. Practically, Eq. (6) permits us to compute the final response $\boldsymbol{r}_T$ at linear time and constant space complexity with respect to training duration, without caching or explicitly computing the batch Hessians (c.f. Eq. (57)). This is akin to forward-mode automatic differentiation for meta-learning [13]. To the best of our knowledge, this is the first application of such techniques to efficient computation of the response. Crucially, if the batch selection $\delta_n^t$ is i.i.d., Eq. (6) defines a *Markov Chain* – an observation that unlocks well-studied mathematical machinery and invites us to analyse its limiting distribution (see Section 4.1).

**Empirical demonstration**. Figure 2 showcases the application of unrolled differentiation as a d-TDA method, successfully predicting changes in the *distribution* of measurements when a select subset of the dataset is removed.

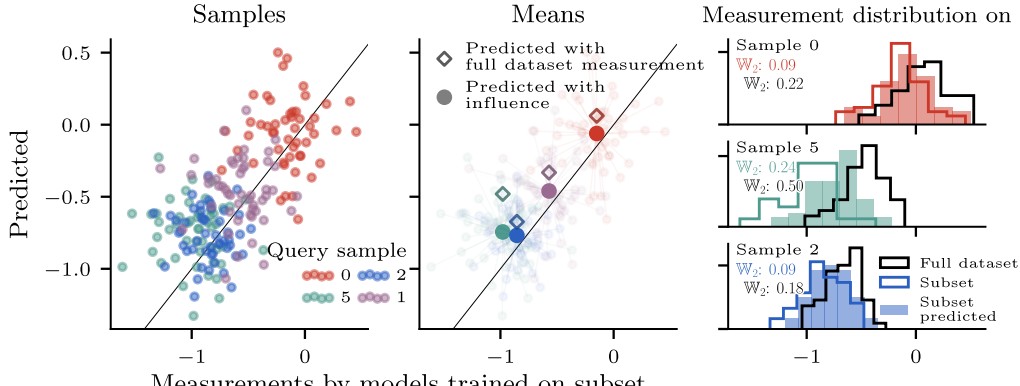

Figure 2: **d-TDA demo for a neural network trained on UCI Concrete**. d-TDA (using unrolled differentiation) gives approximate samples from the distribution of models re-trained on some fixed subset $\mathcal{D}' \subset \mathcal{D}$ *(left)*. Actual samples from $\mu_{\mathcal{D}'}$ (obtained by expensive retraining) are closer to predicted samples from $\mu_{\mathcal{D}'}$ (obtained by efficient d-TDA methods) than they are to samples from the original model $\mu_{\mathcal{D}}$, both in terms of their means *(centre)* and Wasserstein distance *(right)*. The distributions are over measurements on query samples for different stochastic training runs.

**Why not use regular TDA with a fixed seed?** A natural question raised by the challenge of stochasticity is: instead of treating the outcome of training as a random variable, why not simply fix all sources of randomness? Why not just stratify by the random choices like initialisation and data ordering? Naively, this seems to recover a deterministic training algorithm, to which one may apply regular (non-distributional) TDA methods. We refer to this as 'fixed-seed TDA'.

Distributional TDA is often preferable to fixed-seed TDA because many methods, like IFs, more accurately perform the d-TDA task, even when failing at the fixed-seed one. For instance, IFs and unrolled differentiation find local perturbations around the current optimum to predict outcomes of counterfactual retraining. However, even fixing all randomness, chaotic training dynamics can push training into a completely different region of the parameter space. Moreover, when using a fixed batch-size, even tiny changes to the training set size can offset at what iteration each datum appears. This means even fixed-seed trajectories can converge to widely different 'optima' under small dataset perturbations, rendering the shift in the original local optimum inadequate. Later in Section 5, we demonstrate IFs perform better on downstream tasks as a distributional TDA method compared to as a fixed-seed TDA method. This suggests the distributional perspective provides a more accurate picture of how influence functions work.

## 4 What do influence functions sample?

In Section 3, we introduced *distributional* TDA, adopting a rigorous mathematical perspective that accounts for stochasticity in training. Here, we demonstrate how d-TDA relates to *classical* influence functions (IFs; Eq. (3)). From two complementary perspectives, we find that IFs are actually 'secretly distributional', appearing as asymptotic samples in specific d-TDA settings. In stark contrast to usual derivations of IFs, which rely on assumptions that are unrealistic for deep learning [2, 14], we place only mild constraints on $\mathcal{L}(\boldsymbol{\theta})$. All proofs are in Section A.

### 4.1 Perspective 1: unrolled differentiation converges a.s. to influence functions

Adopting a distributional perspective, a natural question is: what is the limiting distribution of the random variable $(\boldsymbol{\theta}_t, \boldsymbol{r}_t)$, updated according to Eq. (6)? Begin by considering the model weights $(\boldsymbol{\theta}_t)$, which are updated by SGD with *i.i.d.* batch selection. We make the following assumptions.

**A1**. $\nabla^2 \mathcal{L}$ and $\nabla \ell_k$ are Lipschitz continuous and bounded.
**A2**. The step sizes $(\eta_t)_{t=0}^{\infty}$ are positive scalars satisfying $\sum_t \eta_t = \infty$ and $\sum_t \eta_t^2 < \infty$.
**A3**. The iterates of Eq. (12) remain bounded a.s., i.e. $\sup_t \|(\boldsymbol{\theta}_t, \boldsymbol{r}_t)\| < \infty$ a.s.

Standard results due to e.g. H. J. Kushner and G. G. Yin [15] give us the following result:

**Theorem 1. (SGD converges to stationary points [15, Theorem 2.1, Chapter 5]).** Provided assumptions A1-A3 hold, the sequence of SGD iterates $(\boldsymbol{\theta}_t)_{t=0}^{\infty}$ as defined by Eq. (6) converges almost surely to the set $\mathcal{S}_{\mathcal{L}}$ of stationary points of the corresponding ODE: $\dot{\boldsymbol{\theta}} = -\nabla \mathcal{L}(\boldsymbol{\theta})$ – namely, $\mathcal{S}_{\mathcal{L}} = \{\boldsymbol{\theta} : -\nabla \mathcal{L}(\boldsymbol{\theta}) = 0\}$.

Theorem 1 demonstrates that, with a suitably decaying learning rate, SGD converges to critical points – namely, saddle points or local minima. Define the set of local minima as follows:

$$\mathcal{S}_{\mathcal{L}}^m := \left\{\boldsymbol{\theta} : -\nabla \mathcal{L}(\boldsymbol{\theta}) = 0, -\nabla^2 \mathcal{L}(\boldsymbol{\theta}) \preceq 0\right\} \subseteq \mathcal{S}_{\mathcal{L}}. \tag{7}$$

If the weights converge to a saddle point in $\mathcal{S}_{\mathcal{L}} \setminus \mathcal{S}_{\mathcal{L}}^m$, it is intuitive that the response $\boldsymbol{r}_t := \frac{\mathrm{d}\boldsymbol{\theta}_t}{\mathrm{d}\varepsilon}\big|_{\varepsilon=0}$ will diverge. This is because the final model parameters will become sensitive to any infinitesimal perturbation of the loss function.[2] Conversely, if the weights converge to a local minimum, the limiting behaviour of $\boldsymbol{r}_t$ becomes tractable. Consider the following additional assumptions.

**A4**. $\nabla \ell_k(\boldsymbol{\theta}) \in \mathrm{Span}(\nabla^2 \mathcal{L}(\boldsymbol{\theta}))$ for all $\boldsymbol{\theta} \in \mathcal{S}_{\mathcal{L}}^m$.
**A5**. The nonzero eigenvalues of $\nabla^2 \mathcal{L}(\boldsymbol{\theta})$ for $\boldsymbol{\theta} \in \mathcal{S}_{\mathcal{L}}^m$ are uniformly bounded away from 0.
**A6**. There exists some compact neighborhood $\mathcal{N}(\mathcal{S}_{\mathcal{L}}^m)$ around $\mathcal{S}_{\mathcal{L}}^m$ such that gradient flow trajectories $\boldsymbol{\theta}(t)$ initialised therein converge uniformly over initialisations to points in $\mathcal{S}_{\mathcal{L}}^m$. Moreover, their lengths are bounded a.s., so that: $\sup_{\boldsymbol{\theta}(0) \in \mathcal{N}(\mathcal{S}_{\mathcal{L}}^m)} \int_{s=0}^{\infty} \|\boldsymbol{\theta}(s) - \lim_{s' \to \infty} \boldsymbol{\theta}(s')\| \mathrm{d}s' < \infty$.

**Theorem 2. (Unrolled differentiation converges to IFs).** Suppose that A1-A6 hold, and consider an SGD trajectory in the set that converges to $\mathcal{S}_{\mathcal{L}}^m$ (c.f. $\mathcal{S}_{\mathcal{L}} \setminus \mathcal{S}_{\mathcal{L}}^m$). The sequence of iterates $((\boldsymbol{\theta}_t, \boldsymbol{r}_t))_{t=0}^{\infty}$ generated by Eq. (6) converges almost surely to the set $\mathcal{R}^* := \{(\boldsymbol{\theta}^*, \boldsymbol{r}_{\mathrm{IF}}(\boldsymbol{\theta}^*) + \boldsymbol{r}_{\mathrm{NS}}(\boldsymbol{\theta}^*)) : \boldsymbol{\theta}^* \in \mathcal{S}_{\mathcal{L}}^m, \boldsymbol{r}_{\mathrm{IF}}(\boldsymbol{\theta}) := \nabla^2 \mathcal{L}(\boldsymbol{\theta})^+ \nabla \ell_k(\boldsymbol{\theta}), \boldsymbol{r}_{\mathrm{NS}}(\boldsymbol{\theta}) \in \mathrm{Null}(\nabla^2 \mathcal{L}(\boldsymbol{\theta}))\}$ – that is, pointwise IFs, plus a component in the Hessian nullspace. $(\cdot)^+$ is the pseudoinverse.

---

[2]This could be interpreted as a limitation of the conventional notions of response and influence.

*Proof sketch.* We consider the ODE which is the continuous time relaxation of Eq. (6). We prove that the solution to this ODE $r(t)$ converges to an influence function, plus a component in the flat directions of a minimum manifold. Under the learning rate assumptions above, the SGD updates asymptotically track this ODE, which allows us to prove the final result. ∎

**Commentary on Theorem 2**. The response iterate $r_t$ either diverges (in the case that the weights $\theta_t$ converge to a saddle), or converges to $\mathcal{R}^*$. Hence, at late times, one can approximately sample from $\mu_{\mathcal{D} \setminus z_k}$ by sampling from $\mu_{\mathcal{D}}$ and offsetting by $\frac{1}{N} r_{\text{IF}}$. Using $r_{\text{IF}}$ instead of $r_{\text{UD}}$ means that 1) we assume we have trained for long enough, and 2) we neglect components of response in the Hessian nullspace. The latter may not converge and will in general depend on the history of SGD iterates $\theta_t$.

**Assumptions**. A1-A3 are standard assumptions, needed to ensure that SGD converges. A4 guarantees the perturbation $\ell_k$ doesn't have a component in the flat directions of the minimum manifold, or else unrolled differentiation diverges. Note that A4 automatically holds for any symmetries shared by $\mathcal{L}$ and $\ell_k$, e.g. due to neural network parameterisation. A5 ensures that IFs remain bounded; Hessian eigenvalues on $\mathcal{S}_{\mathcal{L}}^m$ can be zero, but not nonzero and arbitrarily small. The most technically meaningful assumption is A6, which assumes gradient flow converges sufficiently fast to local minima. We stress that it is much less restrictive than the requirements usually cited for IFs to apply, such as strong convexity [2, 9, 14].

**Remark 2.** For Generalised Linear Models (GLMs), the component of the unrolled response $r_t$ in the nullspace of the Hessian is 0 throughout training. Hence, in this setting, Theorem 2 gives *exact* convergence of the unrolled response to the influence functions formula.

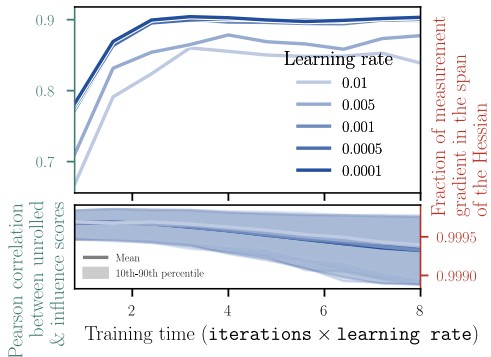

**Empirical validation.** In Figure 3, we test our theoretical results on a regression task with UCI Concrete. As predicted by Theorem 2, the strength of correlation becomes very high (90%) at late times as the unrolled differentiation Markov Chain converges to IFs. As expected, the correlation is better for lower step sizes, for which the SGD iterates (normalised by learning rate) track gradient flow more closely.

Figure 3: **Validating Theorem 2**. *Top:* Correlation between changes in measurement predicted by unrolled differentiation and changes predicted by IFs, plotted against training time. The coefficient becomes high as the Markov chain converges. The correlation is stronger for small $\eta$ where SGD is closer to gradient flow [15]. *Bottom:* Norm of the measurement gradient component in the span of the Hessian divided by norm of the measurement gradient. The nullspace component of $\nabla m$ remains tiny.

We also experimentally confirm that **the component of the unrolled response $r_t$ in the null space of the Hessian** – that is, the error term that IFs cannot capture – **is insignificant for the practical tasks we test**. In the lower panel of Figure 3, we see that the measurement gradient $\nabla m$ only has a tiny component in the nullspace of the Hessian, living almost entirely in the column space. Recalling that $m(\theta^*(\mathcal{D} \setminus z_k)) \approx m(\theta^*(\mathcal{D})) + \frac{1}{N} \nabla m^\top r_{\text{UD}}$, this means it barely contributes to predicted changes in $m$. As such, the fact that IFs do not capture this part of the limiting distribution of $r_t$ does not appear to be of substantial concern for downstream tasks.

### 4.2 Perspective 2: transport maps between Boltzmann distributions

Departing from unrolled differentiation, we now instead model the final weights by a *Boltzmann distribution*. This is motivated by the fact that it is the limiting distribution of Stochastic Gradient Langevin Dynamics [16, 17], which closely resembles SGD. Given an energy function $\mathcal{L}(\theta) - \varepsilon \ell_k(\theta)$ and an inverse temperature parameter $\beta \in \mathbb{R}^+$, the Boltzmann distribution is:

$$p_\varepsilon^\beta(\theta) = \frac{e^{-\beta(\mathcal{L}(\theta) - \varepsilon \ell_k(\theta))}}{Z(\beta, \varepsilon)} \qquad Z(\beta, \varepsilon) = \int e^{-\beta(\mathcal{L}(\theta) - \varepsilon \ell_k(\theta))} \, d\theta. \qquad (8)$$

Let $P_\varepsilon^\beta$ denote the corresponding measure. Let $\mathcal{S}_{\mathcal{L}}^g := \{\theta : \mathcal{L}(\theta) = \inf_{\theta \in \mathbb{R}^d} \mathcal{L}(\theta)\}$ denote the set of *global* minima of the loss function (c.f. $\mathcal{S}_{\mathcal{L}}^m$ above). Consider the following set of assumptions.

**A1**. The derivatives $\frac{d^n \ell_i(\boldsymbol{\theta})}{d\boldsymbol{\theta}^n}$ are bounded for $n \in \{1, 2, 3\}$ and $i \in [\![1, N]\!]$.

**A2**. The nonzero eigenvalues of the Hessian $\nabla^2 \mathcal{L}(\boldsymbol{\theta})$ are uniformly bounded away from zero on $\mathcal{S}_{\mathcal{L}}^g$.

**A3**. The perturbation $\ell_i(\boldsymbol{\theta})$ is constant on $\mathcal{S}_{\mathcal{L}}^g$.

**A4**. $\boldsymbol{\theta} \mapsto \mathcal{L}(\boldsymbol{\theta}) - \varepsilon \ell_k(\boldsymbol{\theta})$ is Lebesgue integrable for all $\varepsilon$ in some neighbourhood of 0.

**A5**. $\mathcal{L}(\boldsymbol{\theta})$ attains its minima, i.e. $\mathcal{S}_{\mathcal{L}}^g = \{\boldsymbol{\theta} : \mathcal{L}(\boldsymbol{\theta}) = \inf_{\boldsymbol{\theta}} \mathcal{L}(\boldsymbol{\theta})\}$ is not empty.

**Theorem 3. (Asymptotic optimality of IFs with Boltzmann distributions).** Let $\mathcal{R} \subset C^1(\mathbb{R}^d, \mathbb{R}^d)$ denote the class of bounded vector fields $\boldsymbol{r}$ such that $T_\varepsilon(\boldsymbol{\theta}) := \boldsymbol{\theta} + \varepsilon \boldsymbol{r}(\boldsymbol{\theta})$ is a $C^1$ diffeomorphism for all sufficiently small $\varepsilon > 0$. Define the functional

$$\mathcal{F}(\boldsymbol{r}, \varepsilon) := \lim_{\beta \to \infty} \frac{1}{\beta} D_{\mathrm{KL}}\left(T_{\varepsilon \#} P_0^\beta | P_\varepsilon^\beta\right), \tag{9}$$

equal to the asymptotic KL divergence between the transformed base measure $T_{\varepsilon \#} P_0^\beta$ and the true perturbed measure $P_\varepsilon^\beta$. Consider the subset of maps $\mathcal{R}_{\mathrm{IF}} := \{\boldsymbol{r} \in \mathcal{R} : \boldsymbol{r}(\boldsymbol{\theta}) = \boldsymbol{r}_{\mathrm{IF}}(\boldsymbol{\theta}) \text{ for } \boldsymbol{\theta} \in \mathcal{S}_{\mathcal{L}}^g\} \subset \mathcal{R}$, for which the map is equal to influence functions on the minimum manifold. Then, given any $\boldsymbol{r} \in \mathcal{R}_{\mathrm{IF}}$ and any $\boldsymbol{r}' \in \mathcal{R} \setminus \mathcal{R}_{\mathrm{IF}}$, there exists some $a \in \mathbb{R}^+$ such that

$$\mathcal{F}(\boldsymbol{r}, \varepsilon) \leq \mathcal{F}(\boldsymbol{r}', \varepsilon) \quad \forall \quad |\varepsilon| \leq a. \tag{10}$$

Moreover, the set $\mathcal{R}_{\mathrm{IF}}$ is non-empty, so such diffeomorphisms do indeed exist.

*Proof sketch*. We start by showing that, for small enough $\varepsilon$, there do indeed exist continuously differentiable bijections in the class $T_\varepsilon(\boldsymbol{\theta})$ such that $\boldsymbol{r}(\boldsymbol{\theta}) = \boldsymbol{r}_{\mathrm{IF}}(\boldsymbol{\theta})$ when $\boldsymbol{\theta} \in \mathcal{S}_{\mathcal{L}}^g$. At low temperatures, only the behaviour at $\mathcal{S}_{\mathcal{L}}^g$ matters because the probability mass concentrates where the loss is minimised. Taking $\beta \to \infty$ and using the Laplace approximation, we analyse the low-temperature KL divergence between $T_{\varepsilon \#} P_0^\beta$ (the transformed measure, without loss perturbation) and $P_\varepsilon^\beta$ (the measure with loss perturbation). Among the class $T_\varepsilon(\boldsymbol{\theta})$, this is minimised at $\mathcal{O}(\varepsilon^2)$ terms by IFs. ∎

**Commentary on Theorem 3**. For Boltzmann distributions, IFs provide *exactly* the transport map in $T_\varepsilon(\boldsymbol{\theta})$ required to transform the low-temperature (weak limit) Boltzmann distribution with loss $\mathcal{L}(\boldsymbol{\theta})$ onto the Boltzmann distribution with a perturbed loss $\mathcal{L}(\boldsymbol{\theta}) - \varepsilon \ell_k(\boldsymbol{\theta})$, up to $\mathcal{O}(\varepsilon^2)$ terms. This provides a very explicit distributional motivation for IFs: they map samples from $P_0^\infty$ (read: $\mu_{\mathcal{D}}$) onto approximate samples from $P_\varepsilon^\infty$ (read: $\mu_{\mathcal{D} \setminus z_k}$), and do so *approximately optimally* in the KL sense. We also remark that, since the KL divergence is invariant under parameter transformation, this notion of optimality does not depend on the specific choice of coordinate system. Minimal assumptions are made on $\mathcal{L}(\boldsymbol{\theta})$ throughout for Theorem 3 to hold.

> **Key takeaways from Section 4**. IFs are implicitly distributional. Supposing $\boldsymbol{\theta}^*(\mathcal{D}) \sim \mu_{\mathcal{D}}$, then the sample $\boldsymbol{\theta}^*(\mathcal{D}) + \frac{1}{N} \boldsymbol{r}_{\mathrm{IF}}$ is approximately distributed according to $\mu_{\mathcal{D} \setminus z_k}$ in two precise mathematical senses: (1) as an asymptotic limit of unrolled differentiation, and (2) minimising a KL divergence if the final weights follow low-temperature Boltzmann distributions. This means that we can use $\boldsymbol{r}_{\mathrm{IF}}$ instead of $\boldsymbol{r}_{\mathrm{UD}}$ in Alg. 1 as a cheaper yet principled proxy, unlocking d-TDA at scale. It may also help explain why IFs are effective in deep learning, far from the convexity assumptions relied upon during typical derivations from robust statistics.

## 5 Distributional Training Data Attribution in Practice

Having demonstrated how d-TDA can be operationalised using IFs (Section 4), we now discuss its practical utility. We begin by demonstrating that distributional influence captures interesting information missing from its classical counterpart.

**Rethinking influence**. Previous papers have heuristically considered what amounts to *mean* influence [12] – if removed from the training dataset, which example would change a model measurement most *on average*? In a synthetic 1D regression task shown in Figure 5, this criterion identifies $x_{30}$ as the most influential datapoint. As discussed above, we could quantify the difference in distributions after retraining in a different way, e.g. with *Wasserstein* influence. Here, in contrast, $x_{31}$ is deemed the most influential. Note that $x_{31}$ is not very influential by conventional measures since its mean shift is modest, yet its removal drastically changes the behaviour after training, sharply increasing uncertainty. This demonstrates that different notions of distributional influence can capture meaningful information about the training data missed by e.g. heuristic ensembling. As a second demonstration, in Figure 11 (App. C.2.3) we use d-TDA to identify MNIST examples which lead to a large change

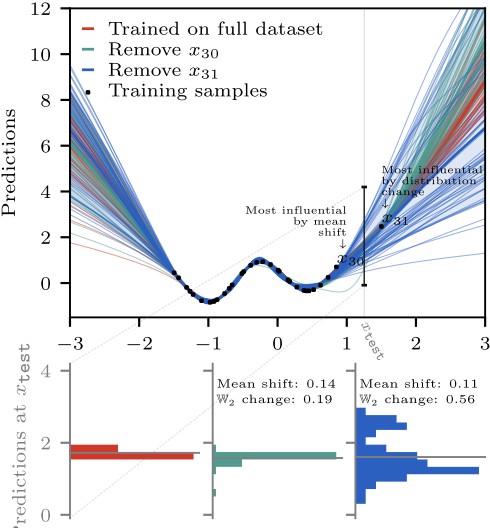

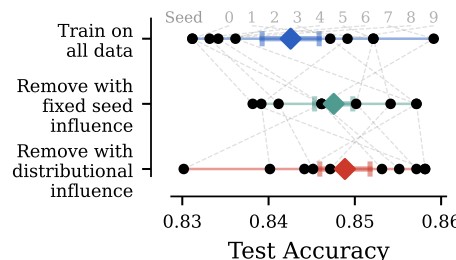

Figure 5: **Distributional influence on 1D regression**. *Top:* Samples of model functions trained on the full dataset, as well as on subsets without the most influential example by mean shift ($x_{30}$) and by Wasserstein shift ($x_{31}$). The 90th percentile of each distribution is indicated by shading. *Bottom:* Histograms of model outputs at $x_{\text{test}}$ after removing $x_{30}$ or $x_{31}$, and corresponding mean and Wasserstein shifts c.f. the original model.

Figure 6: **d-TDA > fixed-seed TDA**. Test accuracy improvements on CIFAR-10 with a SWIN Vision Transformer from IF data pruning. We compare two approaches to subset selection: **1)** traditional TDA with a fixed seed, where for each random seed we remove 5000 datapoints that are predicted to decrease the validation loss the most *for the model trained with that specific fixed seed*; and **2)** distributional-TDA, where for each model we remove 5000 datapoints predicted to decrease the validation loss the most *on average*. Both methods lead to test accuracy improvements upon the baseline trained with all data. However, d-TDA leads to greater overall improvements on average. Black dots show accuracies for individual models (with seeds indicated in gray), whereas coloured diamonds ◆ indicate the average result for each method.

in variance but a small change in mean. The right panel of the figure confirms that retraining does actually lead to the changes our methods predict.

**Distributional influence on diffusion models**. To illustrate this concept at scale, we apply distributional influence to identify the most influential training examples for a latent diffusion model [7]. Figure 7 ranks the most and least influential examples on ArtBench, comparing Wasserstein influence, mean influence, and classical fixed-seed influence. The identified examples vary in each case.

The ability to predict how different training examples impact the distribution over training runs may be practically useful. For example, one could identify examples to add or remove to most reduce variance, hence reducing model (epistemic) uncertainty. Further, d-TDA methods could also be used to operationalise criteria like information gain [18, 19] or marginal likelihood [20, 21, 22, 23].

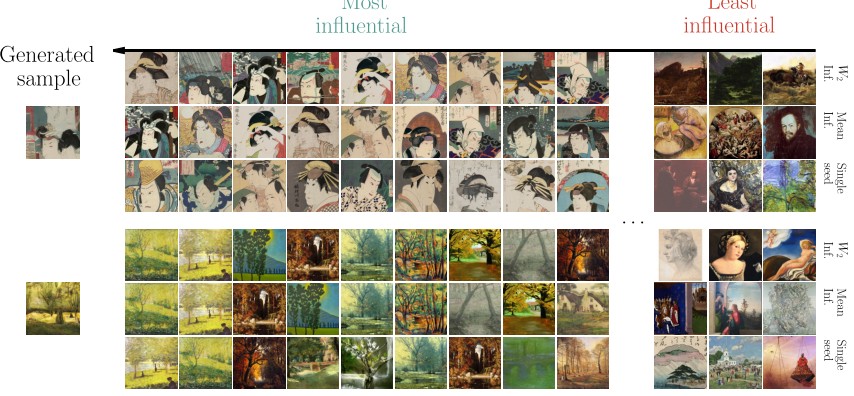

Figure 7: **d-TDA highlights different influences for diffusion models** The figure shows most and least influential datapoints on generations of shown samples from a **latent diffusion model** trained on ArtBench-10. The "most influential" examples are those that change the DDPM loss (a proxy for the log-likelihood) of the generated sample the most, following [7].

**Data pruning with d-TDA**. Next, we apply distributional (mean) influence to a data pruning task, where the goal is to remove datapoints from the training set to improve the performance of the final trained model. We consider a *SWIN transformer* [24] trained on the full `CIFAR-10` dataset (see for details). For the baseline, we remove 5000 datapoints deemed to be most influential – that is, estimated to decrease the validation loss the most when ablated – using regular TDA on a single model. For d-TDA , we remove 5000 datapoints estimated to decrease the validation loss the most *on average*, for 10 models trained using different random seeds. We then compare the final accuracies and losses for *individual* models trained with those examples ablated, using the same random seeds. Figure 6 shows the results. The distributional variant unlocks accuracy gains c.f. fixed-seed TDA.

**Evaluating TDA methods.** The observations above also invite us to rethink how we *evaluate* data attribution methods for stochastic training algorithms: d-TDA methods ought to be effective at identifying examples responsible for large changes in distribution. These changes are often missed when one only looks at the change in mean. Note that the *Linear Datamodelling Score* (**LDS**) [12], a common evaluation metric, can already be interpreted as a d-TDA evaluation metric. It measures how accurately attribution methods rank training datapoints by *mean influence*:

$$\text{LDS} = \text{spearman}\Big[\big(\text{DistInf}_{\boldsymbol{\theta}^*}(\mathcal{D}_i')\big)_{i=1}^M ; \big(\text{DistInf}_{\widetilde{\boldsymbol{\theta}}^*}(\mathcal{D}_i')\big)_{i=1}^M\Big], \quad (11)$$

where spearman denotes the Spearman rank correlation, $\text{DistInf}_{\boldsymbol{\theta}^*}$, $\text{DistInf}_{\widetilde{\boldsymbol{\theta}}^*}$ are distributional (mean) influence scores computed using exact retraining and a d-TDA method respectively, and $\mathcal{D}_i'$ are randomly subsampled subsets of the training data. In light of our discussion, it is natural to generalise Eq. (11) using other notions of distributional influence. We term such metrics **distributional LDS**, of which regular LDS is a special case. We show a preliminary benchmark in Figure 10 (App. C.2), showcasing that distributional LDS can better flesh out differences between d-TDA methods. Distributional LDS (e.g. Wasserstein) can be computed at virtually no additional cost over standard LDS, and we argue should become the default for benchmarking data attribution in deep learning.

**Leave-one-out is not broken, just noisy.** Prior works have reported that TDA methods such as IFs are incapable of accurately predicting the outcome of *leave-one-out* (LOO) retraining, often obtaining near 0% correlation to ground-truth measurements after actual retraining [14]. Adopting a distributional perspective, we view this differently. For big datasets, removing a training example $z_k$ only leads to a tiny change in distribution $\mu_{\mathcal{D}} \rightarrow \mu_{\mathcal{D} \setminus z_k}$. We have seen that IFs allow us to approximately sample from $\mu_{\mathcal{D} \setminus z_k}$, but we may need many empirical samples to detect such a minor distributional shift empirically. In other words, real-world training is noisy; TDA methods struggle with LOO primarily because of a low signal-to-noise ratio, rather than any fundamental incompatibility. In Figure 8, we verify that common attribution methods *are* capable of accurately approximating the LOO distribution with enough samples – the means of the predicted and ground-truth distributions correlate extremely well.

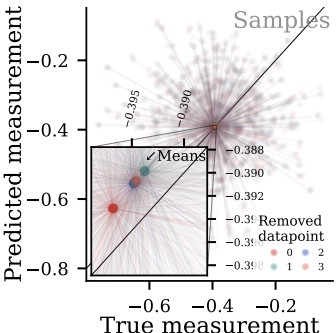

Figure 8: **There is signal in leave-one-out.** Measurements (model output) on a fixed query input when different examples are removed from the training set. True measurements on the $x$-axis against measurements predicted with unrolled differentiation on the $y$-axis for individual models (with different random seeds) are shown with low-opacity. The distributions of measurements are noisy, and similar for each removed example, hence the LOO correlation is close to 0. The empirical *means* of the distribution over random seeds are shown in full color in the inset axis. There is clear correlation between the means of the true and predicted measurements. See App. C.1 for full details.

## 6 Conclusion

This paper introduced distributional training data attribution (d-TDA): a new paradigm for data attribution when training algorithms are stochastic. To demonstrate its utility, we used d-TDA to more effectively identify training examples whose removal improves test loss and accuracy, and proposed novel ways to evaluate d-TDA methods. Rigorously tackling distributional questions also yielded new mathematical motivations for influence functions for deep learning.

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

# A Proofs

## A.1 Proof of Theorem 2: Unrolled differentiation converges to IFs

This appendix provides a proof of Theorem 2, repeated below for the reader's convenience.

Consider stochastic gradient descent updates given by

$$\begin{pmatrix} \boldsymbol{\theta}_{t+1} \\ \boldsymbol{r}_{t+1} \end{pmatrix} = \begin{pmatrix} \boldsymbol{\theta}_t - \frac{\eta_t}{B} \sum_{n=1}^{N} \delta_n^t \nabla \ell_n(\boldsymbol{\theta}_t) \\ \left( I - \frac{\eta_t}{B} \sum_{n=1}^{N} \delta_n^t \nabla^2 \ell_n(\boldsymbol{\theta}_t) \right) \boldsymbol{r}_t + \frac{\eta_t}{B} \delta_k^t \nabla \ell_k(\boldsymbol{\theta}_t) \end{pmatrix}. \tag{12}$$

with random variables $\boldsymbol{\delta}^t$ in $\{0,1\}^N$ for $t \in \mathbb{N}$, *independently and identically distributed* with mean $\mathbb{E}[\delta_n^t] = \frac{B}{N}$, and for some $\boldsymbol{\theta}_0$ independent of $(\boldsymbol{\delta}^t)_{t \in \mathbb{N}}$.

Consider the following assumptions.

**A1**. $\nabla^2 \mathcal{L}$ and $\nabla \ell_k$ are Lipschitz continuous and bounded.
**A2**. The step sizes $(\eta_t)_{t=0}^{\infty}$ are positive scalars satisfying $\sum_t \eta_t = \infty$ and $\sum_t \eta_t^2 < \infty$.
**A3**. The iterates of Eq. (12) remain bounded a.s., i.e. $\sup_t \|(\boldsymbol{\theta}_t, \boldsymbol{r}_t)\| < \infty$ a.s.
**A4**. $\nabla \ell_k(\boldsymbol{\theta}) \in \text{Span}(\nabla^2 \mathcal{L}(\boldsymbol{\theta}))$ for all $\boldsymbol{\theta} \in \mathcal{S}_{\mathcal{L}}^m$.
**A5**. The nonzero eigenvalues of $\nabla^2 \mathcal{L}(\boldsymbol{\theta})$ for $\boldsymbol{\theta} \in \mathcal{S}_{\mathcal{L}}^m$ are uniformly bounded away from 0.
**A6**. There exists some compact neighborhood $\mathcal{N}(\mathcal{S}_{\mathcal{L}}^m)$ around $\mathcal{S}_{\mathcal{L}}^m$ such that gradient flow trajectories $\boldsymbol{\theta}(t)$ initialised therein converge uniformly over initialisations to points in $\mathcal{S}_{\mathcal{L}}^m$. Moreover, their lengths are bounded a.s., so that: $\sup_{\boldsymbol{\theta}(0) \in \mathcal{N}(\mathcal{S}_{\mathcal{L}}^m)} \int_{s=0}^{\infty} \|\boldsymbol{\theta}(s) - \lim_{s' \to \infty} \boldsymbol{\theta}(s')\| \, ds < \infty$.

Define $\mathcal{S}_{\mathcal{L}} := \{\theta : \nabla \mathcal{L}(\boldsymbol{\theta}) = 0\}$, the set of model parameters where the loss function has zero gradient. Also define the set of local *minima*, $\mathcal{S}_{\mathcal{L}}^m := \{\boldsymbol{\theta} : -\nabla \mathcal{L}(\boldsymbol{\theta}) = 0, -\nabla^2 \mathcal{L}(\boldsymbol{\theta}) \preceq 0\} \subseteq \mathcal{S}_{\mathcal{L}}$. We denote the pseudoinverse of a matrix with $(\cdot)^+$. The following is true.

**Theorem A.1. (Unrolled differentiation converges to IFs)**. Suppose that A1-A6 hold, and consider an SGD trajectory in the set that converges to $\mathcal{S}_{\mathcal{L}}^m$ (c.f. $\mathcal{S}_{\mathcal{L}} \setminus \mathcal{S}_{\mathcal{L}}^m$). The sequence of iterates $((\boldsymbol{\theta}_t, \boldsymbol{r}_t))_{t=0}^{\infty}$ generated by Eq. (12) converges almost surely to the set $\mathcal{R}^* := \{(\boldsymbol{\theta}^*, \boldsymbol{r}_{\text{IF}}(\boldsymbol{\theta}^*) + \boldsymbol{r}_{\text{NS}}(\boldsymbol{\theta}^*)) : \boldsymbol{\theta}^* \in \mathcal{S}_{\mathcal{L}}^m, \boldsymbol{r}_{\text{IF}}(\boldsymbol{\theta}) := \nabla^2 \mathcal{L}(\boldsymbol{\theta})^+ \nabla \ell_k(\boldsymbol{\theta}), \boldsymbol{r}_{\text{NS}}(\boldsymbol{\theta}) \in \text{Null}(\nabla^2 \mathcal{L}(\boldsymbol{\theta}))\}$ – that is, pointwise IFs, plus a component in the Hessian nullspace.

*Proof.*

We are interested in the behaviour of response for trajectories where $\boldsymbol{\theta}_t$ converges to $\mathcal{S}_{\mathcal{L}}^m$, rather than $\mathcal{S}_{\mathcal{L}} \setminus \mathcal{S}_{\mathcal{L}}^m$ (see Theorem 1). Consider the following ordinary differential equation:

$$\begin{pmatrix} \dot{\boldsymbol{\theta}} \\ \dot{\boldsymbol{r}} \end{pmatrix} = \begin{pmatrix} -\nabla \mathcal{L}(\boldsymbol{\theta}) \\ -\nabla^2 \mathcal{L}(\boldsymbol{\theta}) \boldsymbol{r} + \nabla \ell_k(\boldsymbol{\theta}) \end{pmatrix}. \tag{13}$$

This can be considered to be a continuous time analogue to the SGD updates in Eq. (12). $\dot{\boldsymbol{\theta}} = -\nabla \mathcal{L}(\boldsymbol{\theta})$ corresponds to gradient flow. We can solve this analytically given the initial model weights $\boldsymbol{\theta}(0)$, obtaining a gradient flow trajectory $\boldsymbol{\theta}(t)$. Note that, by assumption A1 (Lipschitz continuity) and the Picard–Lindelöf theorem, for any initialisation $(\boldsymbol{\theta}(0), \boldsymbol{r}(0))$ there exists a unique solution to the ODE in Eq. (13). The following is true.

**Lemma A.2.** (Gradient Flow ODE converges to influence functions) Given assumptions A1, A4, consider any initialisation $(\boldsymbol{\theta}(0), \boldsymbol{r}(0))$ of Eq. (13) for which **1)** the ODE converges to some limiting weights $\boldsymbol{\theta}^* := \lim_{t \to \infty} \boldsymbol{\theta}(t)$, **2)** the trajectory length is bounded, i.e. $\int_{t=0}^{\infty} \|\boldsymbol{\theta}(t) - \boldsymbol{\theta}^*\| \, dt < \infty$, and **3)** the limiting weights are a (possibly degenerate) local minimum, i.e. $\nabla^2 \mathcal{L}(\boldsymbol{\theta}^*) \succeq 0$. Then $\lim_{t \to \infty} \boldsymbol{r}(t)$ exists and $\lim_{t \to \infty} \boldsymbol{r}(t) \in \{\boldsymbol{r}_{\text{IF}}(\boldsymbol{\theta}^*) + \boldsymbol{r}_{\text{NS}}(\boldsymbol{\theta}^*) : \boldsymbol{r}_{\text{IF}}(\boldsymbol{\theta}) := -\nabla^2 \mathcal{L}(\boldsymbol{\theta})^+ \nabla \ell_k(\boldsymbol{\theta}), \boldsymbol{r}_{\text{NS}}(\boldsymbol{\theta}) \in \text{Null}(\nabla^2 \mathcal{L}(\boldsymbol{\theta}))\}$.

*Proof.* Inserting the computed flow trajectory $\boldsymbol{\theta}(t)$, consider the ODE for influence $\dot{\boldsymbol{r}} = -\nabla^2 \mathcal{L}(\boldsymbol{\theta}(t))\boldsymbol{r} + \nabla \ell_k(\boldsymbol{\theta}(t))$. For notational simplicity and consistency with the dynamical systems literature, we will write this in shorthand as:

$$\dot{\boldsymbol{r}}(t) = \boldsymbol{A}(t)\boldsymbol{r}(t) + \boldsymbol{b}(t), \tag{14}$$

where $\boldsymbol{A}(t) := -\nabla^2 \mathcal{L}(\boldsymbol{\theta}(t))$ and $\boldsymbol{b}(t) := \nabla \ell_k(\boldsymbol{\theta}(t))$. Since $\boldsymbol{\theta}(t)$ converges and the Hessian and gradients are Lipschitz continuous (assumption A1), $\boldsymbol{A}(t) \to \boldsymbol{A}_\infty$ and $\boldsymbol{b}(t) \to \boldsymbol{b}_\infty$ converge as well.

We will now study convergence to the limiting influence, $\boldsymbol{r}_\infty := \lim_{t \to \infty} \boldsymbol{r}(t)$. For a particular flow trajectory with limiting negative Hessian $\boldsymbol{A}_\infty := \lim_{t \to \infty} \boldsymbol{A}(t)$, let $\boldsymbol{P} := \boldsymbol{A}_\infty \boldsymbol{A}_\infty^+$ denote the projection operator onto the column space of $\boldsymbol{A}(t)$. Define the variable $\boldsymbol{x}(t) := \boldsymbol{r}(t) - (-\boldsymbol{A}_\infty^+ \boldsymbol{b}_\infty)$.[3] Rewriting Eq. (14), we have that

$$\dot{\boldsymbol{x}}(t) = \boldsymbol{A}(t)\boldsymbol{x}(t) + \underbrace{(\boldsymbol{b}(t) - \boldsymbol{A}(t)\boldsymbol{A}_\infty^+ \boldsymbol{b}_\infty)}_{:=\boldsymbol{y}(t)}. \tag{15}$$

Note that $\boldsymbol{y}(t) \to 0$ if and only if $\boldsymbol{b}_\infty$ is in the column space of $\boldsymbol{A}_\infty$. Assumption A4 ensures that this is indeed the case. Denote $\boldsymbol{A}(t) = \boldsymbol{A}_\infty + \boldsymbol{\Delta}(t)$, with $\boldsymbol{\Delta}(t) \to 0$. Let $\boldsymbol{x}^\perp(t) := \boldsymbol{P}\boldsymbol{x}(t)$ and $\boldsymbol{x}^\|(t) = (\boldsymbol{I} - \boldsymbol{P})\boldsymbol{x}(t)$ be the components of $\boldsymbol{x}(t)$ in the column and null-space of $\boldsymbol{A}_\infty$ respectively. Our goal will be to show that $\boldsymbol{x}^\perp(t) \to 0$. Premultiplying Eq. (15) by $\boldsymbol{P}$, we have that

$$\dot{\boldsymbol{x}}^\perp(t) = \boldsymbol{A}_\infty \boldsymbol{x}^\perp(t) + \boldsymbol{P}\big(\boldsymbol{\Delta}(t)\big(\boldsymbol{x}^\|(t) + \boldsymbol{x}^\perp(t)\big) + \boldsymbol{y}(t)\big). \tag{16}$$

This implies that

$$(\boldsymbol{x}^\perp)^\top \dot{\boldsymbol{x}}^\perp = (\boldsymbol{x}^\perp)^\top \boldsymbol{A}_\infty \boldsymbol{x}^\perp + (\boldsymbol{x}^\perp)^\top \big(\boldsymbol{\Delta}(\boldsymbol{x}^\| + \boldsymbol{x}^\perp) + \boldsymbol{y}\big), \tag{17}$$

where we suppressed $t$ dependence for compactness. Note that $(\boldsymbol{x}^\perp)^\top \dot{\boldsymbol{x}}^\perp = \frac{1}{2}\frac{d}{dt}\big((\boldsymbol{x}^\perp)^\top \boldsymbol{x}^\perp\big) = \frac{1}{2}\frac{d}{dt}\|\boldsymbol{x}^\perp\|^2 = \|\boldsymbol{x}^\perp\|\frac{d}{dt}\|\boldsymbol{x}^\perp\|$. Also, since $\boldsymbol{x}^\perp$ is in the column space of $\boldsymbol{A}_\infty$ and $\boldsymbol{A}_\infty$ is negative semidefinite, we have that $(\boldsymbol{x}^\perp)^\top \boldsymbol{A}_\infty \boldsymbol{x}^\perp \leq -\lambda \|\boldsymbol{x}^\perp\|^2$ with $-\lambda < 0$ the greatest nonzero eigenvalue of $\boldsymbol{A}_\infty$. Combining the above,

$$\|\boldsymbol{x}^\perp\|\frac{d}{dt}\|\boldsymbol{x}^\perp\| = \|(\boldsymbol{x}^\perp)^\top \boldsymbol{A}_\infty \boldsymbol{x} + (\boldsymbol{x}^\perp)^\top \boldsymbol{P}(\boldsymbol{\Delta}\boldsymbol{x} + \boldsymbol{y})\|$$
$$\leq \|(\boldsymbol{x}^\perp)^\top \boldsymbol{A}_\infty \boldsymbol{x}^\perp(t)\| + \|(\boldsymbol{x}^\perp)^\top \boldsymbol{P}(\boldsymbol{\Delta}\boldsymbol{x} + \boldsymbol{y})\| \leq -\lambda \|\boldsymbol{x}^\perp\|^2 + \|\boldsymbol{x}^\perp\| \|\boldsymbol{P}\| \|\boldsymbol{\Delta}\boldsymbol{x} + \boldsymbol{y}\| \tag{18}$$
$$\leq -\lambda \|\boldsymbol{x}^\perp\|^2 + \|\boldsymbol{x}^\perp\|(\|\boldsymbol{\Delta}\|\|\boldsymbol{x}\| + \|\boldsymbol{y}\|).$$

We used the triangle and Cauchy-Schwarz inequalities. By the assumptions in the theorem statement, $\|\boldsymbol{x}(t)\|$ is bounded by a constant $\gamma$ independent of $t$, which is guaranteed as $\|\boldsymbol{x}(0)\|$ is bounded and $\int_{t=0}^{\infty}\|\boldsymbol{\theta}(t) - \boldsymbol{\theta}(\infty)\|_2 \, dt < \infty$; see Lemma A. 3 below. In this case, we have that

$$\|\boldsymbol{x}^\perp\|\frac{d}{dt}\|\boldsymbol{x}^\perp\| \leq -\lambda \|\boldsymbol{x}^\perp\|^2 + \|\boldsymbol{x}^\perp\|(\gamma\|\boldsymbol{\Delta}\| + \|\boldsymbol{y}\|). \tag{19}$$

Divide through by $\|\boldsymbol{x}^\perp\|$. Since $\boldsymbol{\Delta}(t) \to 0$ and $\boldsymbol{y}(t) \to 0$, for any $\varepsilon > 0$ there exists a time $T_\varepsilon$ such that $\gamma\|\boldsymbol{\Delta}(t)\| + \|\boldsymbol{y}(t)\| \leq \varepsilon$ for all $t > T_\varepsilon$. At such times, $\frac{d}{dt}\big(\|\boldsymbol{x}^\perp(t)\|e^{\lambda t}\big) \leq \varepsilon e^{\lambda t}$, whereupon

$$\|\boldsymbol{x}^\perp(t)\| \leq \|\boldsymbol{x}^\perp(T_\varepsilon)\|e^{-\lambda(t-T_\varepsilon)} + \frac{\varepsilon}{\lambda}\big(1 - e^{-\lambda(t-T_\varepsilon)}\big) \leq \gamma e^{-\lambda(t-T_\varepsilon)} + \frac{\varepsilon}{\lambda}. \tag{20}$$

For any $\varepsilon' > 0$, choosing $\varepsilon$ such that $\frac{\varepsilon}{\lambda} < \varepsilon'$, one can find some $T_{\varepsilon'} > T_\varepsilon$ so that $\|\boldsymbol{x}^\perp(t)\| < \varepsilon'$ for $t > T_{\varepsilon'}$. This proves that $\boldsymbol{x}^\perp(t) \to 0$, whereupon we can conclude that, for such initialisations, $\boldsymbol{P}\boldsymbol{r}(t) \to \boldsymbol{A}_\infty^+ \boldsymbol{b}_\infty$. ∎

As a brief digression: in Lemma A. 2, we used that $\|\boldsymbol{x}(t)\|$ is bounded by some constant $\gamma$. We stated that this is guaranteed if $\|\boldsymbol{x}(0)\|$ is bounded and $\int_{s=0}^{\infty}\|\boldsymbol{\theta}(s) - \boldsymbol{\theta}_\infty\|_2 \, ds < \infty$. This is seen as follows.

**Lemma A.3.** Under assumptions A1-A7 – especially, Lipschitz smoothness of the Hessian and gradients, and the convergence condition $\int_{t=0}^{\infty}\|\boldsymbol{\theta}(t) - \boldsymbol{\theta}_\infty\|_2 \, dt < \infty$ with $\boldsymbol{A}_\infty \preceq 0$ – the response $\boldsymbol{r}(t)$ remains bounded.

*Proof.* Take $\boldsymbol{x}(t) := \boldsymbol{r}(t) - (-\boldsymbol{A}_\infty^+ \boldsymbol{b}_\infty)$. The nonzero eigenvalues of $\boldsymbol{A}_\infty$ are uniformly bounded away from 0 on $\mathcal{S}_\mathcal{L}^m$ and $\boldsymbol{b}_\infty$ is bounded (A4), so bounded $\boldsymbol{x}(t)$ implies bounded $\boldsymbol{r}(t)$. Consider that, for $\dot{\boldsymbol{x}}(t) = \boldsymbol{A}(t)\boldsymbol{x}(t) + \boldsymbol{y}(t)$, we have:

---

[3] $\boldsymbol{x}(t)$ can be interpreted as the error between $\boldsymbol{r}(t)$ and the asymptotic influence functions formula.

$$\|\boldsymbol{x}\|\frac{\mathrm{d}}{\mathrm{d}t}\|\boldsymbol{x}\| = \underbrace{\boldsymbol{x}^\top \boldsymbol{A}_\infty \boldsymbol{x}}_{\leq 0} + \boldsymbol{x}^\top \boldsymbol{\Delta}\boldsymbol{x} + \boldsymbol{x}^\top \boldsymbol{y} \leq \|\boldsymbol{\Delta}\|\|\boldsymbol{x}\|^2 + \|\boldsymbol{x}\|\|\boldsymbol{y}\|. \tag{21}$$

We used the assumption that $\boldsymbol{A}_\infty \preceq 0$, since we are considering flow trajectories that converge to local minima. It follows that $\frac{\mathrm{d}}{\mathrm{d}t}\|\boldsymbol{x}\| \leq \|\boldsymbol{\Delta}\|\|\boldsymbol{x}\| + \|\boldsymbol{y}\|$, so

$$\begin{aligned}
\|\boldsymbol{x}(t)\| &\leq e^{\int_{s=0}^t \|\boldsymbol{\Delta}(s)\|\,\mathrm{d}s}\left(\|\boldsymbol{x}(0)\| + \int_{t'=0}^t \|\boldsymbol{y}(t')\|\, e^{\int_{s'=0}^{t'} -\|\boldsymbol{\Delta}(s')\|\,\mathrm{d}s'}\,\mathrm{d}t'\right)\\
&\leq e^{\int_{s=0}^t \|\boldsymbol{\Delta}(s)\|\,\mathrm{d}s}\left(\|\boldsymbol{x}(0)\| + \int_{t'=0}^t \|\boldsymbol{y}(t')\|\,\mathrm{d}t'\right).
\end{aligned} \tag{22}$$

This is bounded if $\int_{s=0}^\infty \|\boldsymbol{\Delta}(s)\|\,\mathrm{d}s < \infty$ and $\int_{s=0}^\infty \|\boldsymbol{y}(s)\|\,\mathrm{d}s < \infty$. If the first condition holds and $\boldsymbol{b}_\infty$ is in the column space of $\boldsymbol{A}_\infty$ (A4), then from the definition of $\boldsymbol{y}(t)$ the second condition simplifies to $\int_{s=0}^\infty \|\boldsymbol{b}(t) - \boldsymbol{b}_\infty\|\,\mathrm{d}s < \infty$. Under Lipschitz smoothness assumptions for the Hessian and perturbation, these conditions are clearly guaranteed by the convergence rate condition on the model weights $\int_{t=0}^\infty \|\boldsymbol{\theta}(t) - \boldsymbol{\theta}_\infty\|\,\mathrm{d}s < \infty$ as claimed. ∎

Lemma A. 2 proved that, provided $\boldsymbol{r}(0)$ is bounded and the model weights $\boldsymbol{\theta}(t)$ converge to a local minimum, the response vector field $\boldsymbol{r}(t)$ evolving according to the flow ODE converges to influence functions. In particular, we found that $\|\boldsymbol{x}^\perp(t)\| \leq \gamma e^{-\lambda(t - T_{\varepsilon'})} + \frac{\varepsilon'}{\lambda}$, with $\lambda$ the infimum over nonzero eigenvalues of the Hessian at points in $\mathcal{S}_\mathcal{L}^m$ (assumed to be bounded away from 0) and $\gamma$ the maximum possible $\|\boldsymbol{x}(t)\|$ (also bounded given Lemma A. 3). Assuming Lipschitz smoothness and bounded $\boldsymbol{A}(t)$ (assumption A1), $\|\boldsymbol{\Delta}(t)\| \leq L_1\|\boldsymbol{\theta}(t) - \boldsymbol{\theta}_\infty\|$ and $\|\boldsymbol{b}(t)\| \leq L_2\|\boldsymbol{\theta}(t) - \boldsymbol{\theta}_\infty\|$ with $L_1, L_2$ bounded constants. Hence, $T_{\varepsilon'}$ is upper bounded by a constant multipled by the maximum time required to guarantee that $\|\boldsymbol{\theta}(t) - \boldsymbol{\theta}_\infty\| < \varepsilon'$. Therefore, provided A6 holds – i.e. flow trajectories converge uniformly in the neighborhood of the local minimum – $\boldsymbol{Pr}(t)$ initialised therein also converges uniformly. This property will be important later in the proof.

We have seen that the influence ODE converges under mild conditions. Our next task is to use this result to prove the convergence of the influence SGD iterates described by Eq. (12). To do this, we invoke classic arguments made (among others) by V. S. Borkar [25].

We can rewrite Equation (12) in the following way:

$$\begin{pmatrix}\boldsymbol{\theta}_{t+1}\\ \boldsymbol{r}_{t+1}\end{pmatrix} = \begin{pmatrix}\boldsymbol{\theta}_t - \frac{\eta_t}{N}\sum_{n=1}^N \nabla\ell_n + \eta_t M_t\\ \left(I - \frac{\eta_t}{N}\sum_{n=1}^N \nabla^2\ell_n(\boldsymbol{\theta}_t)\right)\boldsymbol{r}_t + \frac{\eta_t}{N}\nabla\ell_k(\boldsymbol{\theta}_t) + \eta_t N_t\end{pmatrix}$$

$$\begin{pmatrix}M_t\\ N_t\end{pmatrix} := \begin{pmatrix}\frac{1}{N}\nabla\mathcal{L}(\boldsymbol{\theta}_t) - \frac{1}{B}\sum_{n=1}^n \delta_n^t \nabla\ell_n(\boldsymbol{\theta}_t)\\ \left[\frac{1}{N}\sum_{n=1}^N \nabla^2\ell_n(\boldsymbol{\theta}_t)\right) - \frac{1}{B}\sum_{n=1}^N \delta_n^t \nabla^2\ell_n(\boldsymbol{\theta}_t)\right]\boldsymbol{r}_t - \left[\frac{1}{N}\nabla\ell_k(\boldsymbol{\theta}_t) - \frac{1}{B}\delta_k^t \nabla\ell_k(\boldsymbol{\theta}_t)\right]\end{pmatrix}.$$

Since the batching variables are i.i.d. and the dataset is fixed, $(M_t, N_t)$ is a Martingale difference sequence with respect to the increasing family of $\sigma$-fields

$$\mathcal{F}_n := \sigma(\boldsymbol{\theta}_m, \boldsymbol{r}_m, m \leq n) \tag{23}$$

That is, $\mathbb{E}((M_{n+1}, N_{n+1})|\mathcal{F}_n) = 0$ a.s., $n \geq 0$. Furthermore, $(M_n, N_n)$ are square integrable with $\mathbb{E}(\|M_{n+1}\|^2 + \|N_{n+1}\|^2 \mid \mathcal{F}_n) \leq K(1 + \|\boldsymbol{\theta}_n\|^2 + \|\boldsymbol{r}_n\|)$ a.s., $n \geq 0$, for some constant $K \geq 0$. This allows us to use standard martingale convergence results to connect the SGD iterates to the ODE solution as $t \to \infty$.

We can think of the SGD trajectories as a noisy discretisation of the corresponding gradient flow ODE, with $t_n := \sum_{k=0}^n \eta_k$ representing the amount of time that the process has been running for. Let $\mathcal{T} := \{t_n : n \in \mathbb{N}\}$ be the corresponding to SGD steps. Let $(\boldsymbol{\theta}_n, \boldsymbol{r}_n)_{n \in \mathbb{N}}$ denote the SGD iterates, generated by Eq. (12). Define $\boldsymbol{r}_{\mathrm{SGD}}(t) := \boldsymbol{r}_n$ for $t \in [t_n, t_{n+1})$. Finally, let $(\boldsymbol{\theta}^m(t), \boldsymbol{r}^m(t))$ for $t \in [t_m, \infty)$ be the solution to the gradient flow ODE in Eq. (13), initialised at $(\boldsymbol{\theta}^m(t_m), \boldsymbol{r}^m(t_m)) = (\boldsymbol{\theta}_m, \boldsymbol{r}_m)$. By [25, Lemma 2.1], since the noise is a martingale difference sequence and given assumptions A5-A6 for any finite $T \in \mathbb{R}^+$:

$$\lim_{m \to \infty} \sup_{t \in [t_m, t_m+T]} \|\boldsymbol{r}_{\text{SGD}}(t) - \boldsymbol{r}^m(t)\| = 0 \quad a.s., \tag{24}$$

so there exists $m_\varepsilon^* \in \mathbb{N}$ such that $\sup_{t \in [t_m+T, t_m+2T]} \|\boldsymbol{r}_{\text{SGD}}(t) - \boldsymbol{r}^m(t)\| \le \varepsilon$ for all $m > m_\varepsilon^*$ [25]. This remains true if we can increase $m_\varepsilon^*$ to be big enough that the time interval $t_m - t_{m-1} < T \, \forall \, m \ge m^*$, whereupon we have that

$$\bigcup_{m : m \ge m^*} [t_m + T, t_m + 2T] = [t_{m^*} + T, \infty). \tag{25}$$

Since $\boldsymbol{\theta}_m \to \mathcal{S}_{\mathcal{L}}^m$, we can make $m_\varepsilon^*$ yet greater to guarantee that the SGD iterates $(\boldsymbol{\theta}_m)_{m \ge m_\varepsilon^*}$ are in the tubular neighborhood where convergence of gradient flow, and therefore the response, is uniform (assumption A6). Recall our earlier definition of the set $\mathcal{R}^* := \{\boldsymbol{r}_{\text{IF}}(\boldsymbol{\theta}^*) + \boldsymbol{r}_{\text{NS}}(\boldsymbol{\theta}^*) : \boldsymbol{\theta}^* \in \mathcal{S}_{\mathcal{L}}^m, \boldsymbol{r}_{\text{IF}}(\boldsymbol{\theta}) := -\nabla^2 \mathcal{L}(\boldsymbol{\theta})^+ \nabla \ell_k(\boldsymbol{\theta}), \boldsymbol{r}_{\text{NS}}(\boldsymbol{\theta}) \in \text{Null}(\nabla^2 \mathcal{L}(\boldsymbol{\theta}))\}$. This corresponds to influence functions at the loss function minima, plus an unspecified component parallel in any degenerate directions. Let $\boldsymbol{P}_m$ denote the unique asymptotic projection operator when gradient flow is initialised at $(\boldsymbol{\theta}_m, \boldsymbol{r}_m)$ and run for infinite time. From the uniform convergence of gradient flow, we have that $\exists T_\varepsilon$ s.t. $\forall t \ge T_\varepsilon$,

$$\inf_{\boldsymbol{r}^* \in R^*} \|\boldsymbol{r}^m(t) - \boldsymbol{r}^*\| \le \inf_{\boldsymbol{r}^* \in R^*} \left( \|\boldsymbol{P}_m \boldsymbol{r}^m(t) - \boldsymbol{P}_m \boldsymbol{r}^*\| + \underbrace{\|(\boldsymbol{I} - \boldsymbol{P}_m)\boldsymbol{r}^m(t) - (\boldsymbol{I} - \boldsymbol{P}_m)\boldsymbol{r}^*\|}_{=0} \right) \tag{26}$$

$$= \inf_{\boldsymbol{r}^* \in R^*} \|\boldsymbol{P}_m \boldsymbol{r}^m(t) - \boldsymbol{P}_m \boldsymbol{r}^*\| \le \varepsilon.$$

The second term vanishes because within the set $\mathcal{R}^*$ the null space component is unconstrained; we make no claims about its convergence. Hence, it can always be exactly fitted to $(\boldsymbol{I} - \boldsymbol{P}_m)\boldsymbol{r}^m(t)$. Meanwhile, the first term is can be made less than $\varepsilon$ due to convergence of $\boldsymbol{r}^m(t)$ perpendicular to flat directions, which we already proved.

Choose any $n$ such that $t_n > t_{m^*} + T_\varepsilon$. Then choose some corresponding $m \ge m^*$ such that $t_n \in [t_m + T_\varepsilon, t_m + 2T_\varepsilon]$, which is always possible due to Eq. (25). Combining the previous inequalities,

$$\inf_{\boldsymbol{r}^* \in R^*} \|\boldsymbol{r}_n - \boldsymbol{r}^*\| \le \underbrace{\|\boldsymbol{r}_n - \boldsymbol{r}^m(t_n)\|}_{\text{SGD} \to \text{ODE}} + \underbrace{\inf_{\boldsymbol{r}^* \in R^*} \|\boldsymbol{r}^m(t_n) - \boldsymbol{r}^*\|}_{\text{ODE} \to \text{influence functions}} \le 2\varepsilon. \tag{27}$$

Take the union over all $t_n > t_{m^*} + T_\varepsilon$, we can finally conclude that

$$\boldsymbol{r}_n \to \mathcal{R}^*, \tag{28}$$

as claimed. This completes the proof. ∎

### A.2 Proof of Theorem 3: Asymptotic optimality of IFs with Boltzmann distributions

This appendix provides a proof of Theorem 3, restated below for convenience.

Given an energy function $\mathcal{L}(\boldsymbol{\theta}) - \varepsilon \ell_k(\boldsymbol{\theta})$ and an inverse temperature parameter $\beta \in \mathbb{R}^+$, the Boltzmann distribution is:

$$p_\varepsilon^\beta(\boldsymbol{\theta}) = \frac{e^{-\beta(\mathcal{L}(\boldsymbol{\theta}) - \varepsilon \ell_k(\boldsymbol{\theta}))}}{Z(\beta, \varepsilon)} \qquad Z(\beta, \varepsilon) = \int e^{-\beta(\mathcal{L}(\boldsymbol{\theta}) - \varepsilon \ell_k(\boldsymbol{\theta}))} \, \mathrm{d}\boldsymbol{\theta}. \tag{29}$$

Let $P_\varepsilon^\beta$ denote the corresponding measure. Let $\mathcal{S}_{\mathcal{L}}^g := \{\boldsymbol{\theta} : \mathcal{L}(\boldsymbol{\theta}) = \inf_{\boldsymbol{\theta} \in \mathbb{R}^d} \mathcal{L}(\boldsymbol{\theta})\}$ denote the set of *global* minima of the loss function (c.f. $\mathcal{S}_{\mathcal{L}}^m$ above). Consider the following set of assumptions.

**A1.** The derivatives $\frac{d^n \ell_i(\boldsymbol{\theta})}{d\boldsymbol{\theta}^n}$ are bounded for $n \in \{1, 2, 3\}$ and $i \in [\![1, N]\!]$.
**A2.** The nonzero eigenvalues of the Hessian $\nabla^2 \mathcal{L}(\boldsymbol{\theta})$ are uniformly bounded away from zero on $\mathcal{S}_{\mathcal{L}}^g$.
**A3.** The perturbation $\ell_i(\boldsymbol{\theta})$ is constant on $\mathcal{S}_{\mathcal{L}}^g$.
**A4.** $\boldsymbol{\theta} \mapsto \mathcal{L}(\boldsymbol{\theta}) - \varepsilon \ell_k(\boldsymbol{\theta})$ is Lebesgue integrable for all $\varepsilon$ in some neighbourhood of 0.
**A5.** $\mathcal{L}(\boldsymbol{\theta})$ attains its minima, i.e. $\mathcal{S}_{\mathcal{L}}^g = \{\boldsymbol{\theta} : \mathcal{L}(\boldsymbol{\theta}) = \inf_{\boldsymbol{\theta}} \mathcal{L}(\boldsymbol{\theta})\}$ is not empty.

**Theorem A.4. (Asymptotic optimality of IFs with Boltzmann distributions.)** Let $\mathcal{R} \subset C^1(\mathbb{R}^d, \mathbb{R}^d)$ denote the class of bounded vector fields $r$ such that $T_\varepsilon(\boldsymbol{\theta}) := \boldsymbol{\theta} + \varepsilon r(\boldsymbol{\theta})$ is a $C^1$ diffeomorphism for all sufficiently small $\varepsilon > 0$. Define the functional

$$\mathcal{F}(r, \varepsilon) := \lim_{\beta \to \infty} \frac{1}{\beta} D_{\mathrm{KL}}\left(T_{\varepsilon \#} P_0^\beta | P_\varepsilon^\beta\right), \tag{30}$$

equal to the asymptotic KL divergence between the transformed base measure $T_{\varepsilon \#} P_0^\beta$ and the true perturbed measure $P_\varepsilon^\beta$. Consider the subset of maps $\mathcal{R}_{\mathrm{IF}} := \{r \in \mathcal{R} : r(\boldsymbol{\theta}) = r_{\mathrm{IF}}(\boldsymbol{\theta}) \text{ for } \boldsymbol{\theta} \in \mathcal{S}_{\mathcal{L}}^g\} \subset \mathcal{R}$, for which the map is equal to influence functions on the minimum manifold. Then, given any $r \in \mathcal{R}_{\mathrm{IF}}$ and any $r' \in \mathcal{R} \setminus \mathcal{R}_{\mathrm{IF}}$, there exists some $a \in \mathbb{R}^+$ such that

$$\mathcal{F}(r, \varepsilon) \leq \mathcal{F}(r', \varepsilon) \quad \forall \quad |\varepsilon| \leq a. \tag{31}$$

Moreover, the set $\mathcal{R}_{\mathrm{IF}}$ is non-empty, so such diffeomorphisms do indeed exist.

*Proof.* We begin with the following lemma.

**Lemma A.5.** For small enough $\varepsilon$, there exist continuously differentiable bijections in the class $T_\varepsilon$ such that $T_\varepsilon(\boldsymbol{\theta}) = \boldsymbol{\theta} + \varepsilon r_{\mathrm{IF}}(\boldsymbol{\theta})$ for $\boldsymbol{\theta} \in S_{\mathcal{L}}$.

*Proof.* We start by defining a function $T_\varepsilon$ that we will show has the claimed properties. Let $\lambda_{\min} \in \mathbb{R}$ denote the smallest nonzero eigenvalue of the Hessian $\nabla^2 \mathcal{L}(\boldsymbol{\theta})$ on the minimum manifold $S_{\mathcal{L}}$, which is bounded away from 0 (assumption A2). Define the following scalar transformation $f : \mathbb{R} \to \mathbb{R}$,

$$f(x) = \begin{cases} -\frac{x}{\lambda_{\min}^2} + \frac{2}{\lambda_{\min}} & \text{if } x < \lambda_{\min}, \\ \frac{1}{x} & \text{otherwise.} \end{cases} \tag{32}$$

Note that $f$ is Lipschitz continuous with constant $1/\lambda_{\min}^2$. For compactness, let $\mathbf{A} := \nabla^2 \mathcal{L}(\boldsymbol{\theta})$. Denote the operation of $f$ on a symmetric matrix $\mathbf{A}$ by $f(\mathbf{A}) := \mathbf{Q}^\top f(\boldsymbol{\Lambda}) \mathbf{Q}$ where $f$ is understood to act separately on each of the eigenvalues on the diagonal of $\boldsymbol{\Lambda}$. Observe that, if $\mathbf{A}$ is positive definite and all its eigenvalues are greater than or equal to $\lambda_{\min}$, then $f(\mathbf{A}) = \mathbf{A}^{-1}$ and we recover the regular matrix inverse. Similarly, if $\mathbf{A}$ is positive semi-definite with all non-zero eigenvalues greater than or equal to $\lambda_{\min}$, and $v$ is a vector in $\mathrm{Span}(\mathbf{A})$, then $\mathbf{A}^+ v = f(\mathbf{A}) v$. Hence, if we take $T_{\varepsilon(\boldsymbol{\theta})} = \boldsymbol{\theta} + \varepsilon r(\varepsilon)$ with $r(\boldsymbol{\theta}) = f(\nabla^2 \mathcal{L}(\boldsymbol{\theta})) \nabla \ell_k(\boldsymbol{\theta})$, this clearly satisfies $r(\boldsymbol{\theta}) = r_{\mathrm{IF}}(\boldsymbol{\theta}) = \nabla^2 \mathcal{L}(\boldsymbol{\theta})^+ \nabla \ell_k(\boldsymbol{\theta})$ for $\boldsymbol{\theta} \in S_{\mathcal{L}}$ (Assumption A.3). Hence, we only need to show that it's a continuous bijection. To this end, we will use the following theorem[4]:

**Lemma A.6.** (Hadamard's Global Inverse Function Theorem S. G. Krantz and H. R. Parks [26, Theorem 6.2.8]) Let $h : \mathbb{R}^d \to \mathbb{R}^d$ be a continuously differentiable function. If:

1. $h$ is proper (for every compact set $K \subset \mathbb{R}^d$, $h^{-1}(K)$ is compact), and
2. the Jacobian of $h$ vanishes nowhere,

then $h$ is a homeomorphism (continuous bijection with a continuous inverse).

We will first show that the Jacobian vanishes nowhere. The Daleckiĭ-Kreĭn Theorem [27, 28] gives us the following:

$$\nabla f(\mathbf{A}) = \mathbf{Q}(\mathbf{R} \odot \mathbf{Q}^\top \nabla \mathbf{A} \mathbf{Q}) \mathbf{Q}^\top, \tag{33}$$

where $\odot$ denotes the *Hadamard matrix product*, $(\mathbf{A} \odot \mathbf{B})_{ij} := \mathbf{A}_{ij} \mathbf{B}_{ij}$, and

$$\mathbf{R}_{ij} = \begin{cases} \frac{f(\lambda_i) - f(\lambda_j)}{\lambda_i - \lambda_j} & \text{if } \lambda_i \neq \lambda_j, \\ f'(\lambda_i) & \text{otherwise.} \end{cases} \tag{34}$$

Note that $\sup_{i,j} |\mathbf{R}_{ij}| \leq \frac{1}{\lambda_{\min}^2}$. The following is true:

$$\|\nabla_i f(\mathbf{A})\|_2 = \|\mathbf{R} \odot \mathbf{Q}^\top \nabla \mathbf{A} \mathbf{Q}\|_2 \leq \sqrt{d_{\mathrm{param}}} \sup_{i,j} |\mathbf{R}_{ij}| \, \|\nabla_i \mathbf{A}\|_2 \leq \sqrt{d_{\mathrm{param}}} \cdot \frac{\|\nabla_i \mathbf{A}\|_{\mathrm{F}}}{\lambda_{\min}^2}. \tag{35}$$

---

[4]Presented for the specific case of the standard topology on $\mathbb{R}^{d_{\mathrm{param}}}$.

Here, $\|\nabla_i \mathbf{A}\|_F$ denotes the Frobenius norm of $\frac{\partial}{\partial \theta_i} \nabla^2 \mathcal{L}(\boldsymbol{\theta})$, which is bounded by a constant if the third derivative of the loss is bounded (assumption A.1).

The Jacobian of this transformation is:

$$\nabla T_\varepsilon(\boldsymbol{\theta}) = \mathbf{I} + \varepsilon \nabla\big(f(\nabla^2 \mathcal{L})\nabla \ell_k\big) = \mathbf{I} + \varepsilon\big[\nabla f(\nabla^2 \mathcal{L})\nabla \ell_k + f(\nabla^2 \mathcal{L})\nabla^2 \ell_k\big], \qquad (36)$$

where $\mathbf{I}$ denotes the $d_{\text{param}} \times d_{\text{param}}$ identity matrix. Given the previous, the spectral radius of the term in square brackets is bounded under assumptions A1-A2, so the Jacobian is positive definite at small enough $\varepsilon$. This means that $T_\varepsilon$ is locally invertible everywhere for small enough $\varepsilon$.

To show that $T_\varepsilon$ is *proper*, note that $T_\varepsilon : \mathbb{R}^{d_{\text{param}}} \to \mathbb{R}^{d_{\text{param}}}$ is proper *iff* $\lim_{n\to\infty} \|T_\varepsilon(\boldsymbol{x}_n)\|_2$ for every sequence $(\boldsymbol{x}_n)_{n=1}^\infty$ s.t. $\|\boldsymbol{x}_n\|_2 \to \infty$ as $n \to \infty$. To show the latter, note that $\nabla^2 \mathcal{L}$ and $\nabla \ell_k$ are bounded (Assumption A.1), and so is $f(\nabla^2 \mathcal{L}(\boldsymbol{\theta}))$ (since $f$ is Lipschitz), and hence $r(\boldsymbol{\theta}) = f(\nabla^2 \mathcal{L})\nabla \ell_k$ is also bounded. Hence, $\lim_{n\to\infty} \| \boldsymbol{x}_n + \varepsilon r(\boldsymbol{x}_n) \|_2 = \infty$ as $\| \boldsymbol{x}_n \|_2 \to \infty$ as required, and by the Hadamard's Theorem, $T_\varepsilon$ is a homeomorphism. ∎

Equipped with Lemma A. 5, for small enough $\varepsilon$ we can apply the change of variables formula. Since the KL-divergence is reparameterisation-invariant, we have that $D_{\text{KL}}\big[T_{\varepsilon\#}P_0^\beta \,\|\, P_\varepsilon^\beta\big] = D_{\text{KL}}\big[T_{\varepsilon\#}^{-1}T_{\varepsilon\#}P_0^\beta \,\|\, T_{\varepsilon\#}^{-1}P_\varepsilon^\beta\big] = D_{\text{KL}}\big[P_0^\beta \,\|\, T_{\varepsilon\#}^{-1}P_\varepsilon^\beta\big]$ for any continuously differentiable bijection $T_\varepsilon$. Note that $T_{\varepsilon\#}^{-1}P_\varepsilon^\beta$ has a density given by

$$p_\varepsilon^\beta(T_\varepsilon(\boldsymbol{\theta}))|\det \nabla T_\varepsilon(\boldsymbol{\theta})| = \frac{e^{-\beta(\mathcal{L}(T_\varepsilon(\boldsymbol{\theta})) - \ell_k(T_\varepsilon(\boldsymbol{\theta})))}}{Z_p(\beta, \varepsilon)} \,|\det \nabla T_\varepsilon(\boldsymbol{\theta})|,$$

Hence, we have that:

$$\frac{1}{\beta}D_{\text{KL}}\big[T_{\varepsilon\#}P_0^\beta \,\|\, P_\varepsilon^\beta\big] = \frac{1}{\beta}D_{\text{KL}}\big[P_0^\beta \,\|\, T_{\varepsilon\#}^{-1}P_\varepsilon^\beta\big]$$

$$= \frac{1}{\beta}\int p_0^\beta(\boldsymbol{\theta}) \log\left(\frac{p_0^\beta(\boldsymbol{\theta})}{p_\varepsilon^\beta(T_\varepsilon(\boldsymbol{\theta})) \,|\det \nabla T_\varepsilon(\boldsymbol{\theta})|}\right) d\boldsymbol{\theta} \qquad (37)$$

$$= \int p_0^\beta(\boldsymbol{\theta})\left(\mathcal{L}(T_\varepsilon(\boldsymbol{\theta})) - \varepsilon\ell_k(T_\varepsilon(\boldsymbol{\theta})) + \frac{1}{\beta}\log\det \nabla T_\varepsilon(\boldsymbol{\theta})\right) d\boldsymbol{\theta} + \frac{1}{\beta}\big(\mathcal{H}\big[p_0^\beta\big] + \log Z_p(\varepsilon, \beta)\big).$$

Here, $\mathcal{H}\big[p_0^\beta\big] := -\int p_0^\beta(\boldsymbol{\theta}) \log p_0^\beta(\boldsymbol{\theta}) \, d\boldsymbol{\theta}$ denotes the entropy of $p_0^\beta$. Note, the terms $\frac{1}{\beta}\big(\mathcal{H}\big[p_0^\beta\big] + \log Z_p(\varepsilon, \beta)\big)$ do not depend on $T_\varepsilon$. Moreover, we have that $\lim_{\beta\to\infty} \frac{1}{\beta}\mathcal{H}\big[p_0^\beta\big] = 0$. Given assumptions A1-A2 used to keep the transformation locally invertible, the eigenvalues of $\nabla T_\varepsilon(\boldsymbol{\theta})$ are bounded by a constant independent of $\boldsymbol{\theta}$ so $\lim_{\beta\to\infty} \frac{1}{\beta}\int p_0^\beta(\boldsymbol{\theta}) \log\det \nabla T_\varepsilon(\boldsymbol{\theta})) = 0$.

Putting in $T_\varepsilon(\boldsymbol{\theta}) = \boldsymbol{\theta} + \varepsilon r(\boldsymbol{\theta})$, let us now consider the expectation $\mathbb{E}_{\boldsymbol{\theta} \sim p_0^\beta}[\mathcal{L}(\boldsymbol{\theta} + \varepsilon r(\boldsymbol{\theta})) - \varepsilon\ell_k(\boldsymbol{\theta} + \varepsilon r(\boldsymbol{\theta}))]$. Below, we will use *Einstein summation notation*, with the implicit understanding that one should sum over repeated indices. Taylor expanding in $\varepsilon$ with $\boldsymbol{\theta}$ fixed,

$$\mathcal{L}(\boldsymbol{\theta} + \varepsilon r) = \mathcal{L}(\boldsymbol{\theta}) + \varepsilon\partial_i\mathcal{L}(\boldsymbol{\theta})r_i + \frac{\varepsilon^2}{2}\partial_{ij}\mathcal{L}(\boldsymbol{\theta})r_i r_j + R_2^{\mathcal{L}}(\boldsymbol{\theta}, \varepsilon, r). \qquad (38)$$

Here, $R_2^{\mathcal{L}}(\boldsymbol{\theta}, \varepsilon, r) = \frac{\varepsilon^3}{3!}\partial_{ijk}\mathcal{L}(\boldsymbol{\theta} + \varepsilon' r)r_i r_j r_k$ is the *error term*,[5] for some $\varepsilon' \in (0, \varepsilon)$. Similarly,

$$\varepsilon\ell_k(\boldsymbol{\theta} + \varepsilon r) = \varepsilon\ell_k(\boldsymbol{\theta}) + \varepsilon^2\partial_i\ell_k(\boldsymbol{\theta})r_i + R_1^{\ell_k}(\boldsymbol{\theta}, \varepsilon, r), \qquad (39)$$

where this time $R_1^{\ell_k}(\boldsymbol{\theta}, \varepsilon, r) = \frac{\varepsilon^3}{2}\partial_{ij}\ell_k(\boldsymbol{\theta} + \varepsilon' r)r_i r_j$. Since the first three derivatives of $\mathcal{L}$ and $\ell_k$, as well as $r$ and $r'$, are bounded a.e. (assumption A1), $\lim_{\varepsilon\to 0} \frac{R_2^{\mathcal{L}}(\boldsymbol{\theta}, \varepsilon, r)}{\varepsilon^3}$ and $\lim_{\varepsilon\to 0} \frac{R_1^{\ell_k}(\boldsymbol{\theta}, \varepsilon, r)}{\varepsilon^3}$ are bounded by constants independent of $\boldsymbol{\theta}$. It follows that

---

[5] With a slight abuse of notation, we included $r$ as an argument to emphasise that the remainder term will depend on the particular choice of function $r$. Once the function $r$ is fixed, the arguments of $R_2^{\mathcal{L}}$ are of course $\boldsymbol{\theta}$ and $\varepsilon$.

$$\mathbb{E}_{\boldsymbol{\theta} \sim p_0^\beta}[\mathcal{L}(\boldsymbol{\theta} + \varepsilon \boldsymbol{r}) - \varepsilon \ell_k(\boldsymbol{\theta} + \varepsilon \boldsymbol{r})]$$

$$= \mathbb{E}_{\boldsymbol{\theta} \sim p_0^\beta}\left[\mathcal{L}(\boldsymbol{\theta}) + \varepsilon \partial_i \mathcal{L}(\boldsymbol{\theta}) \boldsymbol{r}_i + \frac{\varepsilon^2}{2} \partial_i \partial_j \mathcal{L}(\boldsymbol{\theta}) \boldsymbol{r}_i \boldsymbol{r}_j - \varepsilon \ell_k(\boldsymbol{\theta}) - \varepsilon^2 \partial_i \ell_k(\boldsymbol{\theta}) \boldsymbol{r}_i\right] + \mathcal{O}(\varepsilon^3). \tag{40}$$

Next, consider the log partition function, $\log Z_p(\varepsilon, \beta) := \log \int e^{-\beta(\mathcal{L}(\boldsymbol{\theta}) - \varepsilon \ell_k(\boldsymbol{\theta}))} d\boldsymbol{\theta}$. Applying the Laplace approximation [20], we find that:

$$\lim_{\beta \to \infty} \frac{1}{\beta} \log Z_p(\varepsilon, \beta) = - \inf_{\boldsymbol{\theta} \in \mathbb{R}^{d_{\text{param}}}} [\mathcal{L}(\boldsymbol{\theta}) - \varepsilon \ell_k(\boldsymbol{\theta})]. \tag{41}$$

Assembling the various pieces, we then have that:

$$\mathcal{F}(\boldsymbol{r}, \varepsilon) = \lim_{\beta \to \infty} \mathbb{E}_{\boldsymbol{\theta} \sim p_0^\beta}[\mathcal{L}(\boldsymbol{\theta}) - \varepsilon \ell_k(\boldsymbol{\theta})] - \inf_{\boldsymbol{\theta} \in \mathbb{R}^{d_{\text{param}}}} [\mathcal{L}(\boldsymbol{\theta}) - \varepsilon \ell_k(\boldsymbol{\theta})] +$$
$$\varepsilon^2 \lim_{\beta \to \infty} \mathbb{E}_{\boldsymbol{\theta} \sim p_0^\beta}\left[\frac{1}{2} \partial_i \partial_j \mathcal{L}(\boldsymbol{\theta}) \boldsymbol{r}_i \boldsymbol{r}_j - \partial_i \ell_k(\boldsymbol{\theta}) \boldsymbol{r}_i\right] + \mathcal{O}(\varepsilon^3). \tag{42}$$

We dropped the $\lim_{\beta \to \infty} \mathbb{E}_{\boldsymbol{\theta} \sim p_0^\beta}[\partial_i \mathcal{L}(\boldsymbol{\theta}) \boldsymbol{r}_i]$ term since the the weak limit $P_0^\infty$ has support on the minimum manifold $S_\mathcal{L} = \{\boldsymbol{\theta} : \mathcal{L}(\boldsymbol{\theta}) = \inf_{\boldsymbol{\theta} \in \mathbb{R}^{d_{\text{param}}}} \mathcal{L}(\boldsymbol{\theta})\}$, where $\partial_i \mathcal{L}(\boldsymbol{\theta}) = 0$ by definition. Since $\partial_i \mathcal{L}(\boldsymbol{\theta}) \boldsymbol{r}_i$ is continuous and bounded, the limit of the expectations is the expectation under the weak limit.

Let us now consider some $\boldsymbol{r} \in \mathcal{R}_{\text{IF}}$ and some $\boldsymbol{r}' \in \mathcal{R} \setminus \mathcal{R}_{\text{IF}}$. Since $\boldsymbol{r}_{\text{IF}}(\boldsymbol{\theta}) := \nabla^2 \mathcal{L}(\boldsymbol{\theta})^+ \nabla \ell_k(\boldsymbol{\theta})$ directly minimises the square bracket on the second line of Eq. (42) for each $\boldsymbol{\theta} \in S_\mathcal{L}$, we have that

$$\mathcal{F}(\boldsymbol{r}, \varepsilon) - \mathcal{F}(\boldsymbol{r}', \varepsilon) =$$
$$\underbrace{\varepsilon^2 \lim_{\beta \to \infty} \mathbb{E}_{\boldsymbol{\theta} \sim p_0^\beta}\left[\frac{1}{2} \partial_i \partial_j \mathcal{L}(\boldsymbol{\theta}) \boldsymbol{r}_i \boldsymbol{r}_j - \partial_i \ell_k(\boldsymbol{\theta}) \boldsymbol{r}_i - \left(\frac{1}{2} \partial_i \partial_j \mathcal{L}(\boldsymbol{\theta}) \boldsymbol{r}'_i \boldsymbol{r}'_j - \partial_i \ell_k(\boldsymbol{\theta}) \boldsymbol{r}'_i\right)\right]}_{:=-\Delta < 0}$$
$$+\varepsilon^3 \lim_{\beta \to \infty} \mathbb{E}_{\boldsymbol{\theta} \sim p_0^\beta}\left[R_2^{\mathcal{L}}(\boldsymbol{\theta}, \varepsilon, \boldsymbol{r}) - R_1^{\ell_k}(\boldsymbol{\theta}, \varepsilon, \boldsymbol{r}) - R_2^{\mathcal{L}'}(\boldsymbol{\theta}, \varepsilon, \boldsymbol{r}') + R_1^{\ell_k'}(\boldsymbol{\theta}, \varepsilon, \boldsymbol{r}')\right]. \tag{43}$$

Every remainder term is bounded by a constant independent of $\boldsymbol{\theta}$ and $\varepsilon$, so the magnitude of the expectation on the bottom line is bounded by a constant $C \in \mathbb{R}^+$. Hence,

$$\mathcal{F}(\boldsymbol{r}, \varepsilon) - \mathcal{F}(\boldsymbol{r}', \varepsilon) \leq -\Delta \varepsilon^2 + C \varepsilon^3, \tag{44}$$

whereupon $\mathcal{F}(\boldsymbol{r}, \varepsilon) \leq \mathcal{F}(\boldsymbol{r}', \varepsilon)$ is guaranteed for $|\varepsilon| \leq \frac{\Delta}{C}$. This completes the proof. ∎

**Extra remark**. In the special case that the perturbation does not introduce any new global minima/ break the degeneracy of the minimum manifold, then the KL divergence actually *vanishes* up to $\mathcal{O}(\varepsilon^3)$ so we have an even stronger result. Assume that the perturbation is constant on $S_\mathcal{L}$ (assumption A3). Consider the following:

$$\inf_{\boldsymbol{\theta} \in \mathbb{R}^d} [\mathcal{L}(\boldsymbol{\theta}) - \varepsilon \ell_k(\boldsymbol{\theta})] =$$
$$\mathcal{L}(\boldsymbol{\theta}^*) - \varepsilon \ell_k(\boldsymbol{\theta}^*) + \varepsilon^2 \inf_{\boldsymbol{\theta}^* \in S_\mathcal{L}}\left[\frac{1}{2} \partial_i \partial_j \mathcal{L}(\boldsymbol{\theta}^*) \boldsymbol{r}_{\text{IF}}(\boldsymbol{\theta}^*)_i \boldsymbol{r}_{\text{IF}}(\boldsymbol{\theta}^*)_j - \partial_i \ell_k(\boldsymbol{\theta}^*) \boldsymbol{r}_{\text{IF}}(\boldsymbol{\theta}^*)_i\right] + \mathcal{O}(\varepsilon^3) \tag{45}$$
$$= \mathcal{L}(\boldsymbol{\theta}^*) - \varepsilon \ell_k(\boldsymbol{\theta}^*) - \frac{1}{2} \varepsilon^2 \inf_{\boldsymbol{\theta}^* \in S_\mathcal{L}}\left[\nabla \ell_k(\boldsymbol{\theta}^*)^\top \nabla^2 \mathcal{L}(\boldsymbol{\theta}^*)^+ \nabla \ell_k(\boldsymbol{\theta}^*)\right] + \mathcal{O}(\varepsilon^3),$$

where we Taylor expanded in $\varepsilon$ and used the implicit function theorem. The infimum will cancel with the expectation on the lower line of Eq. (42) if its argument $\nabla \ell_k(\boldsymbol{\theta}^*)^\top \nabla^2 \mathcal{L}(\boldsymbol{\theta}^*)^+ \nabla \ell_k(\boldsymbol{\theta}^*)$ is identical for all $\boldsymbol{\theta}^* \in S_\mathcal{L}$. This will not be true for general perturbations – just a specific class that does not break the manifold symmetry at $\mathcal{O}(\varepsilon^2)$.

To provide an intuitive summary: influence functions are always the best local transport map at $\mathcal{O}(\varepsilon^2)$, parameterised by $\boldsymbol{\theta} \to \boldsymbol{\theta} + \varepsilon \boldsymbol{r}(\boldsymbol{\theta})$. But they are also the best non-local map, making $\mathcal{O}(\varepsilon^2)$ terms vanish so that the KL divergence is truly $\mathcal{O}(\varepsilon^3)$, in the special case that the perturbation does not induce symmetry breaking of the minimum manifold at $\mathcal{O}(\varepsilon^2)$. This condition is formalised by

$\nabla \ell_k(\boldsymbol{\theta}^*)^\top \nabla^2 \mathcal{L}(\boldsymbol{\theta}^*)^+ \nabla \ell_k(\boldsymbol{\theta}^*)$ being constant for all $\boldsymbol{\theta}^* \in S_{\mathcal{L}}$ – which is also intuitive because it gives the change in $\ell_k$ at the new minima.

# B Derivation of Influence Functions

The purpose of this appendix section is to provide a standalone "classical" derivation of the influence functions framework for the "classical" training data attribution task. We state the Implicit Function Theorem (Section B.1); then, in Section B.2 we introduce the details of how it can be applied to predict local changes in the minima of a loss function $\mathcal{L}(\varepsilon, \boldsymbol{\theta})$ parameterised by a continuous hyperparameter $\varepsilon$ (e.g. $\mathcal{L}(\varepsilon, \boldsymbol{\theta}) = \mathcal{L}_{\mathcal{D}}(\boldsymbol{\theta}) - \varepsilon \ell_k(\boldsymbol{\theta})$), so that $\varepsilon$ controls how down-weighted the loss terms on some examples are). This derivation largely mirrors that in [7, Appendix A].

There appears to be a prevalent misconception that influence functions can only be applied to convex loss functions [14]. This appendix hopefully makes clear that they can be applied to a loss function with multiple minima, as long as each minimum is a strict local minimum; influence functions in that case will simply predict the change in the corresponding local minimum. The rest of this paper then makes formal what influence functions do in the more general and complex setting of stochastic optimisation for general loss functions with possibly degenerate minima.

## B.1 The Implicit Function Theorem

*Theorem 1 (Implicit Function Theorem S. G. Krantz and H. R. Parks)*: Let $F : \mathbb{R}^n \times \mathbb{R}^m \to \mathbb{R}^m$ be a continuously differentiable function, and let $\mathbb{R}^n \times \mathbb{R}^m$ have coordinates $(\boldsymbol{x}, \boldsymbol{y})$. Fix a point $(\boldsymbol{a}, \boldsymbol{b}) = (a_1, ..., a_n, b_1, ..., b_m)$ with $F(\boldsymbol{a}, \boldsymbol{b}) = \boldsymbol{0}$, where $\boldsymbol{0} \in \mathbb{R}^m$ is the zero vector. If the Jacobian matrix $\nabla_{\boldsymbol{y}} F(\boldsymbol{a}, \boldsymbol{b}) \in \mathbb{R}^{m \times m}$ of $\boldsymbol{y} \mapsto F(\boldsymbol{a}, \boldsymbol{y})$, defined as

$$\left[\nabla_{\boldsymbol{y}} F(\boldsymbol{a}, \boldsymbol{b})\right]_{ij} := \frac{\partial F_i}{\partial y_j}(\boldsymbol{a}, \boldsymbol{b}), \tag{46}$$

is invertible, then there exists an open set $U \subset \mathbb{R}^n$ containing $\boldsymbol{a}$ such that there exists a unique function $g : U \to \mathbb{R}^m$ satisfying $\boldsymbol{g}(\boldsymbol{a}) = \boldsymbol{b}$, and $F(\boldsymbol{x}, \boldsymbol{g}(\boldsymbol{x})) = \boldsymbol{0}$ for all $\boldsymbol{x} \in U$. Moreover, $g$ is continuously differentiable.

*Remark 1 (Derivative of the Implicit Function)*: Denoting the Jacobian matrix of $\boldsymbol{x} \mapsto F(\boldsymbol{x}, \boldsymbol{y})$ as $\nabla_{\boldsymbol{x}} F(\boldsymbol{x}, \boldsymbol{y})$, the derivative $\frac{\partial \boldsymbol{g}}{\partial \boldsymbol{x}} : U \to \mathbb{R}^{m \times n}$ of $\boldsymbol{g}$ given by Theorem 1 can be written as:

$$\frac{\partial \boldsymbol{g}}{\partial \boldsymbol{x}} = -\left[\nabla_{\boldsymbol{y}} F(\boldsymbol{x}, \boldsymbol{g}(\boldsymbol{x}))\right]^{-1} \nabla_{\boldsymbol{x}} F(\boldsymbol{x}, \boldsymbol{g}(\boldsymbol{x}))]. \tag{47}$$

This can readily be seen by noting that, for $\boldsymbol{x} \in U$:

$$F(\boldsymbol{x}', \boldsymbol{g}(\boldsymbol{x}')) = \boldsymbol{0} \quad \forall \boldsymbol{x}' \in U \quad \Rightarrow \quad \frac{\mathrm{d} F(\boldsymbol{x}, \boldsymbol{g}(\boldsymbol{x}))}{\mathrm{d} \boldsymbol{x}} = \boldsymbol{0}. \tag{48}$$

Since $g$ is differentiable (by Theorem 1), we can apply the chain rule of differentiation to get:

$$\boldsymbol{0} = \frac{\mathrm{d} F(\boldsymbol{x}, \boldsymbol{g}(\boldsymbol{x}))}{\mathrm{d} \boldsymbol{x}} = \nabla_{\boldsymbol{x}} F(\boldsymbol{x}, \boldsymbol{g}(\boldsymbol{x})) + \nabla_{\boldsymbol{y}} F(\boldsymbol{x}, \boldsymbol{g}(\boldsymbol{x})) \frac{\partial \boldsymbol{g}(\boldsymbol{x})}{\partial \boldsymbol{x}}. \tag{49}$$

Rearranging gives equation (47).

## B.2 Applying the implicit function theorem to quantify the change in the optimum of a loss

Consider a loss function $\mathcal{L} : \mathbb{R}^n \times \mathbb{R}^m \to \mathbb{R}$ that depends on some hyperparameter $\varepsilon \in \mathbb{R}^n$ (e.g. the scalar by which certain loss terms are down-weighted) and some parameters $\boldsymbol{\theta} \in \mathbb{R}^m$. At the minimum of the loss function $\mathcal{L}(\varepsilon, \boldsymbol{\theta})$, the derivative with respect to the parameters $\boldsymbol{\theta}$ will be zero. Hence, assuming that the loss function is twice continuously differentiable (hence $\frac{\partial L}{\partial \varepsilon}$ is continuously differentiable), and assuming that for some $\varepsilon' \in \mathbb{R}^n$ we have a set of parameters $\boldsymbol{\theta}^\star$ such that $\frac{\partial \mathcal{L}}{\partial \varepsilon}(\varepsilon', \boldsymbol{\theta}^\star) = \boldsymbol{0}$ and the Hessian $\frac{\partial^2 \mathcal{L}}{\partial \boldsymbol{\theta}^2}(\varepsilon', \boldsymbol{\theta}^\star)$ is invertible, we can apply the implicit function theorem

to the derivative of the loss function $\frac{\partial \mathcal{L}}{\partial \varepsilon} : \mathbb{R}^n \times \mathbb{R}^m \to \mathbb{R}^m$, to get the existence of a continuously differentiable function $g$ such that $\frac{\partial \mathcal{L}}{\partial \varepsilon}(\varepsilon, g(\varepsilon)) = \mathbf{0}$ for $\varepsilon$ in some neighbourhood of $\varepsilon'$.

Now $g(\varepsilon)$ might not necessarily be a minimum of $\boldsymbol{\theta} \mapsto \mathcal{L}(\varepsilon, \boldsymbol{\theta})$. However, by making the further assumption that $\mathcal{L}$ is strictly convex we can ensure that whenever $\frac{\partial \mathcal{L}}{\partial \boldsymbol{\theta}}(\varepsilon, \boldsymbol{\theta}) = \mathbf{0}$, $\boldsymbol{\theta}$ is a unique minimum, and so $g\{\varepsilon\}$ represents the change in the minimum as we vary bold$\{\varepsilon\}$. Alternatively, if $\boldsymbol{\theta}^\star = g(\varepsilon')$ is a local minimum, then $g(\varepsilon)$ will give the shift in this particular local minimum as we vary $\varepsilon$ in some neighbourhood around $\varepsilon'$.

We can make this more precise with the following lemma:

*Lemma 1*: Let $\mathcal{L} : \mathbb{R}^n \times \mathbb{R}^m \to \mathbb{R}$ be a twice continuously differentiable function, with coordinates denoted by $(\varepsilon, \boldsymbol{\theta}) \in \mathbb{R}^n \times \mathbb{R}^m$, such that $\boldsymbol{\theta} \mapsto \mathcal{L}(\varepsilon, \boldsymbol{\theta}))$ is strictly convex $\forall \varepsilon \in \mathbb{R}^n$. Fix a point $(\varepsilon', \boldsymbol{\theta}^\star)$ such that $\frac{\partial \mathcal{L}}{\partial \boldsymbol{\theta}}(\varepsilon', \boldsymbol{\theta}^\star) = \mathbf{0}$. Then, by the Implicit Function Theorem applied to $\frac{\partial \mathcal{L}}{\partial \boldsymbol{\theta}}$, there exists an open set $U \subseteq \mathbb{R}^n$ containing $\boldsymbol{\theta}^\star$ and a unique function $g : U \to \mathbb{R}^m$ satisfying:
- $g(\varepsilon') = \boldsymbol{\theta}^\star$, and
- $g(\varepsilon)$ is the unique minimum of $\boldsymbol{\theta} \mapsto \mathcal{L}(\varepsilon, \boldsymbol{\theta})$ for all $\varepsilon \in U$.

Moreover, $g$ is continuously differentiable with derivative:

$$\frac{\partial g(\varepsilon)}{\partial \varepsilon} = - \left[ \frac{\partial^2 \mathcal{L}}{\partial \boldsymbol{\theta}^2}(\varepsilon, g(\varepsilon)) \right]^{-1} \frac{\partial^2 \mathcal{L}}{\partial \varepsilon \partial \boldsymbol{\theta}}(\varepsilon, g(\varepsilon)) \tag{50}$$

Again, dropping the assumption of strict convexity, and replacing it with the assumption that $(\varepsilon', \boldsymbol{\theta})$ merely yield a local minimum, gives a similar conclusion, but only guarantees existence of a function $g$ such that $g(\varepsilon)$ is a *local* minimum for all $\varepsilon \in U$.

Equation (50) might still look a bit distinct from the influence function formula. The one missing piece is restricting ourselves to look at $\mathcal{L}$ of the form $\mathcal{L}(\varepsilon, \boldsymbol{\theta}) = \mathcal{L}_{\mathcal{D}}(\boldsymbol{\theta}) - \varepsilon \ell(\boldsymbol{\theta})$, matching the loss interpolations we consider in the main paper body.

*Remark 2*: For a loss function $\mathcal{L} : \mathbb{R} \times \mathbb{R}^m$ of the form $\mathcal{L}(\varepsilon, \boldsymbol{\theta}) = \mathcal{L}_{\mathcal{D}}(\boldsymbol{\theta}) - \varepsilon \ell(\boldsymbol{\theta})$, $\frac{\partial^2 \mathcal{L}}{\partial \varepsilon \partial \boldsymbol{\theta}}(\varepsilon, g(\varepsilon))$ in the equation above simplifies to:

$$\frac{\partial^2 \mathcal{L}}{\partial \varepsilon \partial \boldsymbol{\theta}}(\varepsilon, g(\varepsilon)) = -\frac{\partial \ell}{\partial \boldsymbol{\theta}}(g(\varepsilon)) \tag{51}$$

The above give the final influence functions formula. Namely, for the loss of the form:

$$\mathcal{L}(\varepsilon, \boldsymbol{\theta}) = \underbrace{\frac{1}{N} \sum_{i=1}^{N} \ell_i(\boldsymbol{\theta})}_{\mathcal{L}_{\mathcal{D}}} - \underbrace{\frac{1}{M} \sum_{j=1}^{M} \ell_{i_j}(\boldsymbol{\theta}) \varepsilon}_{\ell} \tag{52}$$

we can substitute $\frac{\partial^2 \mathcal{L}}{\partial \varepsilon \partial \boldsymbol{\theta}} = -\frac{1}{M} \sum_{j=1}^{M} \frac{\partial}{\partial \boldsymbol{\theta}} \ell_{i_j}(\boldsymbol{\theta})$ into (50) to get the existence of a function $g$ with the properties given by Lemma 1 with the derivative taking the following familiar form:

$$\frac{\partial g(\varepsilon)}{\partial \varepsilon} = \left[ \frac{\partial^2 \mathcal{L}}{\partial \boldsymbol{\theta}^2}(\varepsilon, g(\varepsilon)) \right]^{-1} \frac{1}{M} \sum_{j=1}^{M} \frac{\partial}{\partial \boldsymbol{\theta}} \ell_{i_j}(\boldsymbol{\theta}), \tag{53}$$

and, at $\varepsilon = 0$:

$$\frac{\partial g}{\partial \varepsilon}(0) = \left[ \frac{\partial^2 \mathcal{L}_{\mathcal{D}}}{\partial \boldsymbol{\theta}^2}(g(0)) \right]^{-1} \frac{1}{M} \sum_{j=1}^{M} \frac{\partial}{\partial \boldsymbol{\theta}} \ell_{i_j}(\boldsymbol{\theta}). \tag{54}$$

## C Additional experimental results

### C.1 Investigating leave-one-out

In Figure 9, we show that d-TDA methods *are* able to approximate the leave-one-out (LOO) distribution over measurements rather well. We simply need a very large number of samples from each distribution to get a good empirical estimate of the distribution in order to observe this. The setting for the LOO experiment was training an MLP on the UCI Concrete dataset (see Section E.1 for architectural and training details). We plot the measurement (model output) for a fixed query input for 500 models trained with different random seeds. For each one of 4 removed (leave-one-out) training examples, we retrain a model with each random seed without that example to obtain the ground-truth measurement shown on the $x$-axis (left & middle plots in Figure 9). To obtain the 'predicted' measurement, we compute the predicted change to the measurement of the model trained on all the data using (exact) unrolled differentiation; this is shown on the y-axis (left & middle plots in Figure 9).

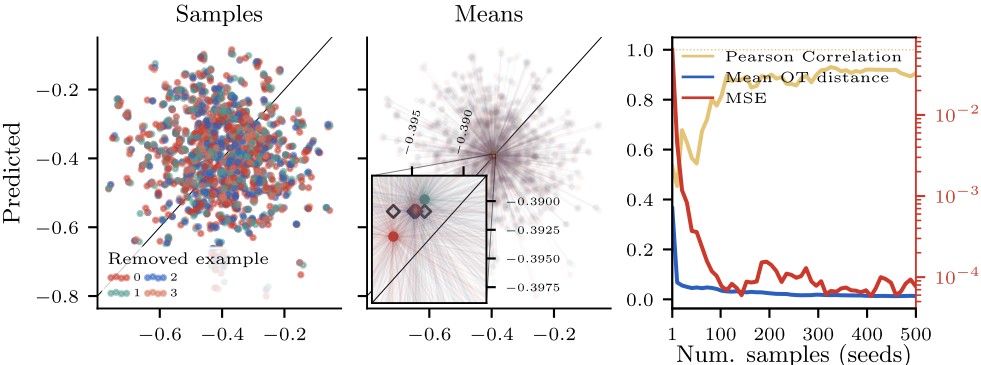

Measurement by models trained on different leave-one-out subsets

Figure 9: **There is signal in leave-one-out.** *Left:* Measurements (model output) on a fixed query example predicted by a d-TDA method (unrolled differentiation) when different singular examples are to be removed from the training set against the actual measurements on models retrained without those examples. The distributions of measurements are noisy, and very similar for each removed example, hence the LOO correlation is close to 0. *Middle:* If we look at the means of the distributions, we see that the measurement distributions *are* subtly different, and the d-TDA method is able to pick up on the shift in mean. The differences are tiny, however; note the scale on the zoomed-in plot. For reference, the mean of the measurement distribution with model trained on the full dataset is shown (on the y-axis) with rectangles; we see that the d-TDA method improves upon using the mean of the original distribution. *Right:* Correlation between the means of the true measurement distributions after retraining, and the means of the predicted distributions, against the number of seeds we use to empirically estimate each distribution. As the number of seeds goes up into the hundreds, the correlation approaches 90%. The seeds (determining data ordering and initialisation) were chosen independently for the fully trained model and the retrained models, indicating that we don't need to correlate the retraining trajectories with the fully trained models to get good LOO scores [10].

### C.2 Distributional Linear Datamodelling Score (LDS)

In Figure 10, we show distributional LDS scores using different notions of distributional influence. It can be seen that different distributional LDS metrics do reveal differences between methods that can't be seen when only using the mean influence. For instance, the performance difference when using the full Hessian vs. block-diagonal Hessian for influence functions only becomes apparent when using the Wasserstein LDS metric. Similarly, EK-FAC with and without score normalisation (described in Section C.2.2) perform identically (up to numerical accuracy) on mean influence LDS, but we see that the normalisation helps slightly when using the Wasserstein and variance change influence LDS metrics. Lastly, it's clear that all methods except for unrolled differentiation fall short of being able to capture the variance change in the measurement distributions in the settings considered.

For future work, we would strongly recommend using the Wasserstein LDS metric in benchmarks. It's a natural choice from a theoretical standpoint – Wasserstein distance is able to capture differences in distributions that go beyond changes in mean. It also makes sense intuitively as a notion of influence in stochastic training settings. Lastly, it's easy to implement — it differs only marginally

from the mean LDS metric — and empirically seems to capture interesting information missed by mean LDS.

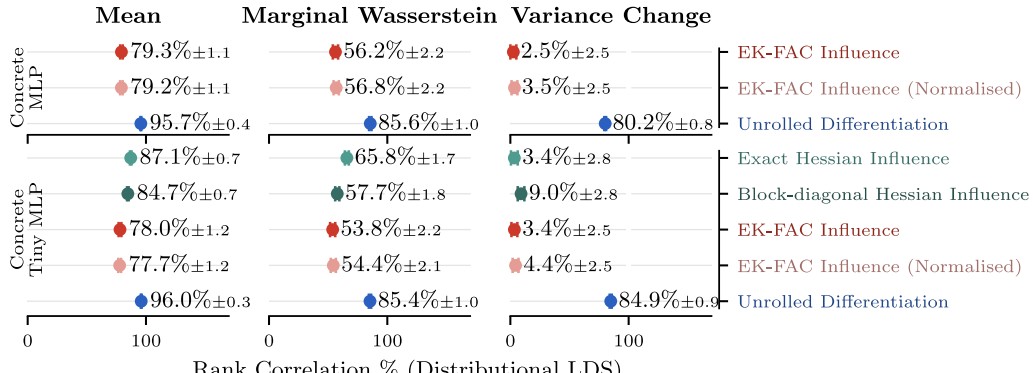

Figure 10: **Distributional LDS.** The distributional LDS scores using different notions of *distributional influence* (mean, variance change and Wasserstein). Each axis row corresponds to a different training setting, and each nested row to a different d-TDA method.

### C.2.1 Influence rankings are different according to different notions of influence

Using distributional LDS with notions of influence other than mean influence would be in vain if they produced the same orderings over (groups of) datapoints. We observe, however, that this is not the case: Wasserstein influence and variance influence produce meaningfully different rankings, presenting a different challenge for d-TDA methods. This is demonstrated in Table 1 below.

Table 1: Similarity between influence rankings for random subsets of the training dataset when using different notions of *distributional influence*. The distributional influence (and the corresponding rankings) are empirically estimated by retraining. "*Top 10% overlap*" refers to the fraction of the 10% most influential subsets that is shared by rankings according to different notions of influence. "*Footrule distance*" represents the total number of places each element in one ordering would have to be shifted by to match the other ordering. The reported footrule distances and top 10% overlaps are the average over all query points

| | **\|Mean\| vs. Wasserstein influence** | | **\|Mean\| vs. Variance influence** | |
| **Setting** | Footrule distance | Top 10% overlap | Footrule distance | Top 10% overlap |
| Concrete \| MLP | 27.4 (max 200) | 47% | 129.7 (max 200) | 8% |
| MNIST \| MLP | 32.2 (max 200) | 54% | 106.3 (max 200) | 11% |

### C.2.2 Normalised Hessian-approximations for influence functions

One previously observed issue when using Hessian approximations such as K-FAC with influence functions is that, although the correlation to ground-truth measurements is good, the scale in the predicted change is often off by a large factor [7]. This deficiency is not captured when looking at classic (mean) LDS metric, as the metric is invariant to the scale of the predicted change in measurement. However, the scale of the predicted change matters when we rank subsets according to influence using other notions of difference in the distribution. Hence, distributional LDS metric can detect methods that are off by a large scale factor in their predictions.

To alleviate this limitation of influence functions on distributional influence tasks, we propose a method to empirically normalise the Hessian approximation. Concretely, we do so **in a way that doesn't require any retraining**, unlike hyperparameter sweeps done to maximise an LDS score.

Concretely, we note that for a Hessian approximation $\widetilde{H}$ to the Hessian $H$, for any vector $v$ in column space of the Hessian, we would want:

$$\| \, \widetilde{\boldsymbol{H}}^+ \boldsymbol{H}\boldsymbol{v} - \boldsymbol{v} \, \|_2^2 \approx 0. \tag{55}$$

If the Hessian approximation is a good approximation to the Hessian, but is off by some scale factor, i.e. $\alpha\widetilde{\boldsymbol{H}} \approx \boldsymbol{H}$ for some $\alpha$, we can find $\alpha$ by trying to minimise:

$$\sum_{\boldsymbol{v}_i} \| \, \alpha\widetilde{\boldsymbol{H}}^+ \boldsymbol{H}\boldsymbol{v}_i - \boldsymbol{v}_i \, \|_2, \tag{56}$$

for a set of vectors $\boldsymbol{v}_i$ that we expect to be in the column space of the Hessian. This is exactly our proposed normalisation method. For the set of vectors $\boldsymbol{v}_i$, we use the per-datapoint training loss gradients, as we would expect them to be in the column space of the true Hessian (otherwise, the response would diverge as training goes on, as shown in our theory section). We can compute $\boldsymbol{H}\boldsymbol{v}_i$ — a Hessian-vector product — relatively cheaply even for large models, at roughly the cost of a forward-backward pass, by using `torch.func.hvp`. Lastly, Eq. (55) is a second-degree polynomial in $\alpha$, and so can be solved analytically. Hence, we don't need to run optimisation to find the normalisation factor $\alpha$. At the end, we simply multiply the Hessian approximation by the normalisation factor $\alpha$ to get the normalised Hessian approximation. We see minor improvements in the distributional LDS scores from using the normalisation factor, but we observe that the normalisation factor is necessary for the predicted changes in distribution by EK-FAC influence to look visually reasonable.

### C.2.3 Identifying examples responsible for high-variance on MNIST

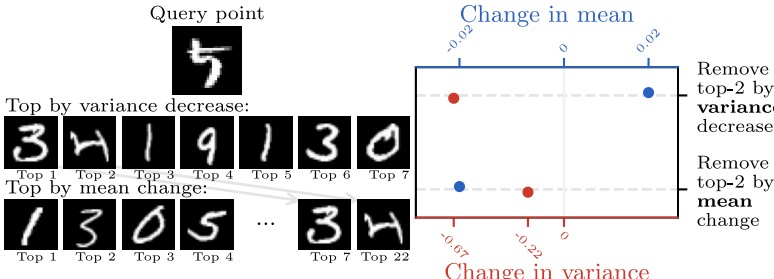

Figure 11: **d-TDA for MNIST**. d-TDA with influence functions (see Section 4) successfully determines which training examples to remove for a decrease in measurement variance. These differ to examples identified for a change in mean. Different d-TDA variants capture diverse information about the training data. This experiment uses a multi-layer perceptron (MLP) trained on MNIST.

### C.2.4 Data Pruning Results

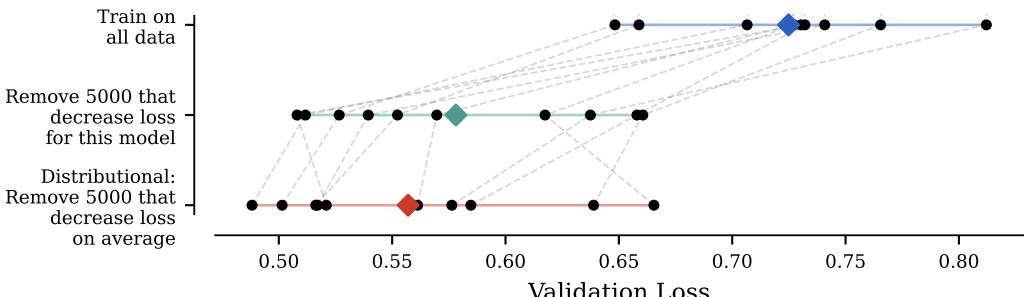

Figure 12: Validation loss improvements on `CIFAR-10` with a SWIN Vision Transformer from IF data pruning. For matching results for accuracy see Figure 6. We compare two approaches to subset selection: **1)** traditional TDA with a fixed seed, where for each random seed we remove 5000 datapoints that are predicted to decrease the validation loss the most for the model trained with that fixed seed; and **2)** distributional-TDA, where for each model we remove 5000 datapoints predicted to decrease the validation loss the most *on average*. Both methods lead to validation loss improvements upon the baseline trained with all data, but d-TDA leads to a greater average improvement. Black dots show accuracies for individual models (with seeds indicated in gray), whereas coloured diamonds indicate ◆ the average result for each method.

## D Related Work

**Influence Functions** Influence functions were originally proposed as a method for data attribution in deep learning by [2]. Later, [29] explored influence functions for investigating the effect of removing or adding groups of data points. Further extensions were proposed by [30] — who explored utilising higher-order information — and [31], who aimed to improve influence ranking via re-normalisation (different from our normalisation). Initial works on influence functions [2, 29] relied on using iterative solvers to compute the required inverse-Hessian-vector products. [3] later explored using EK-FAC as an alternative solution to efficiently approximate calculations with the inverse Hessian. [14] investigated the empirical limitations of influence functions for predicting changes in measurements in the leave-one-out setting, without taking into consideration the distributional aspects of the training algorithm. [11] also investigated the limitations of influence functions, and propose perspectives on what if not counterfactual retraining they might actually approximate. In this paper, we propose alternative perspectives, which are truthful to the underlying goal of predicting outcomes of coutnerfactual retraining with data removed.

**Unrolled differentiation** Orthogonally, pointing out the limitations of influence functions, [8, 9, 10] have proposed to use *unrolled differentiation* for computing influence instead. For SGD trajectories, one can apply the chain rule of differentiation to obtain a closed-form formula for the unrolled differentiation response:

$$\frac{\mathrm{d}\boldsymbol{\theta}_T(\varepsilon)}{\mathrm{d}\varepsilon}\bigg|_{\varepsilon=0} = -\sum_{t=0}^{T-1}\delta_t^k\left(\prod_{l=t}^{T-2}\left(I - \frac{\eta_l}{B}\sum_{i=1}^{N}\delta_i^l\nabla^2\ell_{z_i}(\boldsymbol{\theta}_l)\right)\right)\frac{\eta_t}{B}\nabla\ell_{z_k}(\boldsymbol{\theta}_t) =: \boldsymbol{r}_{\text{UD}}. \quad (57)$$

**K-FAC and EK-FAC** The need for approximate computation with the training loss Hessian in deep learning is evident, and Kronecker-Factored Approximate Curvature (K-FAC) has been one of the best performing Hessian approximations in TDA that can be run on a large scale. K-FAC was originally proposed by [32] to approximate the Fisher Information matrix for natural gradient descent. It was initially formulated only for multi-layer perceptrons, but has since been generalised to any architecture with linear layers with weight-sharing (which includes convolutional neural networks, recurrent neural networks, and transformers) by [33]. [34] introduced eigenvalue-corrected K-FAC (EK-FAC), which corrects K-FAC by using the optimal diagonal approximation in the Kronecker-factored eigenbasis. This was originally done in the context of approximate natural gradient descent, but [3] later used EK-FAC in the influence function approximation to study generalisation in large language models.

# E Experimental Details

## E.1 Settings

We work with the following training settings in the empirical investigations in this paper:

**Concrete | MLP**. In this setting, we train a multi-layer perceptron (MLP) on a (1D target) regression setting on the UCI Concrete dataset [35]. The MLP with an input size of 8, hidden dimensions of `[128, 128, 128]`, and GeLU activation functions, was trained using Stochastic Gradient Descent (SGD) with a learning rate of 0.03 and momentum of 0.9. We applied a weight decay of $10^{-5}$ and gradient clipping at 1.0. The model was trained for 580 iterations using a mean squared error (MSE) loss function and a batch size of 32. The initial 58 iterations (10% of the total) are dedicated to a linear learning rate warmup from 0. For all the retrained models with the data removed, we keep the same number of training iterations as the original model, no matter how much data is removed.[6]

**Concrete | Tiny MLP**. This setting is the same as the previous one, but we use a smaller MLP with hidden dimensions of `[64, 64]`, which enables exact Hessian inversion.

**MNIST | MLP**. For the MNIST | MLP setting, we train a multi-layer perceptron (MLP) on the MNIST dataset [36]. The MLP takes flattened $28 \times 28$ images (input size 784), has hidden dimensions of `[512, 256, 128]`, and an output size of 10. The model was trained using SGD with a learning rate of 0.03 and momentum of 0.9. We applied a weight decay of $10^{-3}$. The model was trained for 1560 iterations with a cross-entropy loss function and a batch size of 64. A linear learning rate warmup from 0 was applied for the initial 5% of the total iterations.

**SWIN Vision Transformer | CIFAR-10**. We train a SWIN Transformer as described in [24] with 2 sets of blocks with 4 layers each, with channel dimensionality $128 \rightarrow 128$ and $128 \rightarrow 256$ respectively. We use attention head dimension of 32, with a patch-size of 2, window-size of 4, and 30% dropout applied to the final layer (head). We train the model with AdamW with a learning rate of $10^{-4}$, weight decay of $10^{-1}$, linear warmup for the first 5% of training iterations, a cosine schedule, and a total of 200000 training iterations with a batch-size of 64. For the dataset, we use the full 50000 images from the train set of `CIFAR-10`.

**Latent Diffusion Model | ArtBench**. For training the model, we follow the `ArtBench` setting with a Latent Diffusion Model as described in Appendix J in B. K. Mlodozeniec, R. Eschenhagen, J. Bae, A. Immer, D. Krueger, and R. E. Turner [7]. The only difference is that we train 5 models with different random seeds on the full dataset.

## E.2 Influence computation

For influence functions computation, we rely on the following methods:

**Exact Hessian** We use `curvlinops` [37] to compute the exact Hessian for the training loss. We add a damping factor equivalent to weight-decay, so that the Hessian corresponds to the actual training loss with the $\ell_2$ penalty.[7]

**Block-diagonal Hessian** We compute the full exact Hessian as described above, but then extract the per-parameter (weights and biases of each layer) block-diagonal parts of the Hessian. The inverse of a block-diagonal matrix is the block-diagonal matrix of inverses of each block, which allows us to invert the Hessian block for each parameter separately. For this, we use the same solver and settings as for the exact Hessian.

**EK-FAC** Eigenvalue-corrected Kronecker-Factored Curvature (EK-FAC) [32, 33, 38] can be viewed as a Kronecker-factored approximation to the Hessian. We use the `curvlinops` [37] implementation of EK-FAC, with the slight modification to compute the *pseudo-inverse* rather than regular inverse, as our theory suggests we ought to do. This amounts to thresholding eigenvalues, and only inverting

---

[6]This is so that the "trajectory length" will be roughly equivalent for trained and retrained models.

[7]Note that, without the $\ell_2$ penalty term, the Hessian will not in general be positive semi-definite when training has converged. Dealing with weight-decay is a tacit detail that is not often mentioned in the literature. For inversion, we use the default `pytorch` pseudo-inverse solver `torch.linalg.pinv` with relative tolerance of $10^{-4}$ and absolute tolerance of 0.

the ones above a certain threshold, while setting the ones below it to zero. This is because, due to numerical errors, 0 eigenvalues might actually be compute as very small values, which after inversion will dominate the inverse matrix spectrum. The threshold is set relative to the largest eigenvalue for each layer, and we set it to $10^{\{-4\}}$ times the largest eigenvalue by default. Just as for the exact Hessian, before taking the pseudo-inverse, we add a damping factor equivalent to weight-decay, so that the Hessian corresponds to the actual training loss with the $\ell_2$ penalty.

**Unrolled differentiation** We compute *exact* unrolled differentiation with forward-mode automatic differentiation, by keeping track of the $\frac{d\boldsymbol{\theta}_t}{d\varepsilon}$ terms during training, and computing the forward derivative through the optimiser update using forward-mode autodiff (`torch.func.jvp`). There is one such term for every datapoint (or group of datapoints considered) to be removed. In the case of stateful optimisers (like SGD with momentum or Adam), we also need to keep track of the derivative of the optimisier state at iteration $t$ with respect to $\varepsilon$.

### E.3 Individual experiment details

**Figure 2**. For this figure, we train 50 MLP models on the UCI Concrete (see Section E.1 for architecture and training details). We remove a fixed randomly sampled subset of 10% of the training datapoints to obtain $\mathcal{D}'$ for the 'retrained' models. We measure and plot the 1D model output on the left and center plot. The 'predicted' measurements are computed using exact unrolled differentiation applied to the models trained on the full dataset.

**Figure 3**. To investigate the correlation between unrolled differentiation and exact Hessian influence functions, we restrict ourselves to the 'Tiny MLP' UCI Concrete setting described in Section E.1. This smaller setting allows us to compute the exact Hessian. For the top axis, we compute changes in measurement (model output) on 103 test points from UCI Concrete, and plot the Pearson correlation between the changes predicted by unrolled differentiation and exact Hessian influence. We also computed the correlation between exact Hessian influence functions and unrolled differentiation with changes to parameters project to lie within the span of the Hessian (assuming eigenvalues below relative tolerance are 0). The lines were virtually overlapping with the original correlation to unrolled differentiation without the projection, and hence removing the null-space component doesn't affect the results significantly.

For the bottom axis, we compute the predicted changes in measurement (model output) for 103 different test points using (1) unrolled differentiation and (2) unrolled differentiation, but projecting the predicted change in parameters onto the span of the Hessian (again, assuming anything below the relative tolerance of $10^{-4}$ is a 0 eigenvalue). We report the Pearson correlation across the 103 test points between measurement changes computed with (1) and (2).

**Transformer Data Pruning (Figure 6 & Figure 12)**. For the data pruning task, we train 10 models with 10 different random seeds on the full `CIFAR-10`. We compute influence functions using EK-FAC [3, 38] with an adaptation to the EK-FAC implementation in the `curvlinops` library [37]. Concretely, we use a numerically stable pseudo-inverse as described in Section E.2. We then find 5000 datapoints to remove for each method in the following way:

- **Fixed-seed TDA**. For each of the 10 models trained with different random seeds, we influence functions to identify *different* 5000 datapoints to remove for each model. We select the 5000 datapoints that are predicted to reduce the validation loss when removed the most *for that model*. When we retrain with the datapoints removed, we use the same random seed as for training the original full model.

- **Distributional (mean influence) TDA**. We identify *one* subset of 5000 datapoints to remove for all 10 models. Concretely, we find 5000 datapoints that are predicted to reduce the the validation loss the most on average over the 10 models. We then retrain each of the 10 models with that same subset removed.

If influence functions performed perfectly as classical TDA methods, identifying a *separate* subset to remove for each random seed should perform at least as well as picking one shared subset for all seeds. The fact that the latter performs better implies that influence functions are indeed better understood (and more accurate) as a d-TDA method.

As a baseline, we compare against the 10 models trained on the full dataset. Naturally, this outperforms removing 5000 datapoints at random, and hence is a stronger baseline.

**Distributional Influence for Latent Diffusion Models (Figure 7)**. To identify top influences for the latent diffusion model, we again use EK-FAC influence, this time without the numerically stable pseudo-inverse. Instead, we directly use the reference implementation open-sourced in [7], and use the same IF settings with damping as described in the appendix of [7] for the `ArtBench` setting. We compute distributional influence with 5 models trained with 5 different random seeds. For the fixed-seed TDA reference, we apply influence functions to one model only (with seed 0) and report the top influences for that seed.

## E.4 Distributional LDS experiments

For the distributional LDS experiments in Section C.2, we subsample 20 datasets from the original dataset uniformly at random without replacement, each with 10% of the datapoints removed (c.f. 100 datasets and 50% examples removed in [12]). For each subdataset, we train 50 models with different seeds. To estimate the distribution of the fully trained model measurements, we also use 50 seeds, and we apply the d-TDA method to produce an approximate sample from the models trained on subdatasets to each of the 50 fully trained models. To estimate mean, variance change and Wasserstein influences, we compute the mean differences, variance differences and Wasserstein distances for the *empirical* distributions of the measurements using the 50 seeds.

