# OpenReview forum: "Distributional Training Data Attribution: What do Influence Functions Sample?"
_NeurIPS.cc/2025/Conference — NeurIPS 2025 spotlight_

### Official Review · Reviewer_A7Ts · 2025-06-25

**Clarity:** 3
**Significance:** 3
**Originality:** 4
**Rating:** 5
**Confidence:** 5

**Summary:**

This paper introduces a new framework called distributional training data attribution for evaluating the influence of training data on machine learning models, particularly in the context of stochastic training methods.
The authors argue that traditional training data attribution methods are insufficient because they do not account for the randomness inherent in deep learning model training, which arises from factors like random weight initialization and stochastic batching.
This randomness means training on the same dataset can produce different models.
The core idea of d-TDA is to shift the focus from predicting a single, deterministic change in model output to predicting how the entire distribution of model outputs changes when a training example is removed or altered.
This approach allows for a more comprehensive understanding of influence, capturing effects on not just the mean outcome but also its variance or other distributional properties.
Key contributions:
1. The paper formally defines the problem of data attribution in stochastic settings as predicting changes in the distribution of model weights or outputs.
This new concept quantifies a training point's importance based on how its removal affects the output distribution, using metrics like the Wasserstein distance.
The paper demonstrates that this can identify influential examples missed by traditional mean-based methods.
2. The paper shows how unrolled differentiation can be used to estimate changes in the output distribution without the need for expensive retraining.
3. The paper rigorously shows that classical influence functions, a popular but poorly understood tool in deep learning, naturally emerge from the d-TDA framework under certain assumptions.
It proves that IFs can be seen as a long-training-time limit of unrolled differentiation and as an optimal way to map between Boltzmann distributions representing the model before and after data removal.
This provides a new theoretical grounding for their effectiveness in non-convex settings.

**Questions:**

See weaknesses

**Ethical Concerns:**

["NO or VERY MINOR ethics concerns only"]

**Final Justification:**

I thank the authors for their comprehensive rebuttal.

The newly added experiments on Vision Transformers and diffusion models strengthen the paper.

The clarifications on the theoretical assumptions and the relationship between Theorems 2 and 3 are also well-received.

I would like to maintain my positive score.

**Limitations:**

yes

**Quality:**

3

**Strengths And Weaknesses:**

Strengths:
1. The primary strength of this paper is its introduction of d-TDA. This is a fundamentally new and important way to think about data attribution that directly confronts the stochasticity inherent in modern machine learning.
Instead of treating randomness as a nuisance to be averaged away, the authors reframe it as a central object of study.
This conceptual shift from predicting a deterministic change to predicting a change in the distribution of model outputs is a significant and timely contribution to the field.

2. The d-TDA framework allows for a much richer understanding of influence.
A key finding is that some data points have a minimal effect on the mean of a model's output but can drastically alter its variance or other distributional properties.
This is demonstrated effectively in a 1D regression task where the example most influential by Wasserstein shift was different from the one most influential by mean shift, and its removal sharply increased prediction uncertainty.
This ability to identify variance-inducing examples is a powerful new tool for debugging models, improving their stability, and understanding generalization.

3. A major contribution is providing a new, rigorous mathematical motivation for the surprising effectiveness of influence functions in deep learning.
The authors show from two complementary perspectives that IFs emerge naturally from the distributional framework.
These derivations are significant because they do not require the restrictive convexity assumptions of traditional IF theory, making them far more applicable to non-convex deep learning models.

Weaknesses:
1. The paper claims that its framework, particularly the use of IFs, unlocks d-TDA for large-scale models like diffusion and language models. However, the experiments presented are on smaller-scale problems (UCI datasets, MNIST with an MLP).
While the efficient forward-mode computation of the unrolled response is a notable contribution to efficiency, the paper does not empirically demonstrate that the full d-TDA approach is computationally feasible or provides meaningful insights on the massive models where attribution methods are arguably most needed.

2. The theoretical results, particularly Theorem 2 regarding the convergence to IFs, rely on a set of assumptions (A1-A6).
While the authors argue these are less restrictive than prior work, their applicability to all practical deep learning scenarios could be questioned.
For example, A6 assumes that gradient flow trajectories converge "sufficiently fast" to local minima.
The extent to which the complex, high-dimensional, and often chaotic dynamics of large neural network training satisfy these conditions remains an open question.
The paper acknowledges and discusses these assumptions, which is good practice, but they remain a limitation on the proven generality of the claims.

3. The paper presents two distinct mathematical motivations for IFs, one based on the limit of SGD dynamics and the other on transport maps between Boltzmann distributions.
While both are elegant and converge on the same result, the paper does not explicitly discuss the relationship between them.
Are they simply parallel justifications, or is there a deeper connection between the long-term behavior of SGD and the properties of the Boltzmann distribution that this work implicitly relies on?
Clarifying this would strengthen the theoretical narrative.

Missing citations:
1. The journey, not the destination: How data guides diffusion models. Arxiv 2024
2. Intriguing Properties of Data Attribution on Diffusion Models. ICLR 2024

---

> ### Author Rebuttal · Authors · 2025-07-31
>
> We thank the reviewer for their detailed comments and praise of the paper. We are happy to address their minor concerns below. We also politely draw their attention to our **extra experiments**, added in response to their comments.
>
> **Applicability to transformers&diffusion models**. The reviewer correctly notes that our chief contributions are conceptual and theoretical, introducing d-TDA, explaining how it can be operationalised with unrolled differentiation and IFs, and rigorously motivating IFs in a deep learning context within this framework with Thms 2 & 3. Whilst we provide a preliminary empirical exploration (see especially Figs 2, 3, 4 and 7), a full study of our methods’ applicability to larger models like LLMs must be left to future work.
>
> That said, we do agree that our messaging might be clearer with experiments on different, bigger architectures. For this reason, we have added **additional experiments applying our d-TDA methods to transformers and diffusion models**.
> For the vision transformer experiments, we show that distributional (mean) influence can better identify training datapoints that will improve the final test loss _fo a single model_ when removed from the training set. By removing 10% of datapoints, we can improve the test loss from 0.715 to 0.509 (compare to 0.528 for regular TDA with a fixed seed). For latent diffusion trained from scratch on a dataset of 50000 Artbench images (256x256 resolution), we compare top predicted Wasserstein shift and mean shift samples, illustrating that they affect the ground-truth retraining distribution in the predicted manner. Thanks for prompting this; we hope this addition strengthens the paper in the reviewer's eyes.
>
> **Assumptions.** The reviewer is right to observe that the assumptions used for our theoretical work are much less restrictive than those commonly used to derive IFs in the literature (namely, convexity). They are also very standard for theoretical work involving SGD. Whilst we do not claim that they will always be universally applicable for deep learning, we suggest that they are actually remarkably general. For instance, bounded loss gradient assumptions could be enforced by considering optimisation on a bounded domain, which is the case in practice when using finite-precision floating-point numbers. We also emphasise that A1-A6 should be interpreted as sufficient conditions for unrolled differentiation to converge to IFs. There could be pathological situations where e.g. SGD itself does not converge (in which cases, deep learning training wouldn't perform), in which case our theoretical results will certainly not hold. We respectfully suggest that this should not be construed as a weakness of our analysis – it is more of a reflection of the mathematical character of the problem.
>
> **Relationship between Thms 2 and 3**. Thanks for the question. These are parallel justifications with some relation to one-another. Roughly speaking, the relationship is that, after running SGD with a small learning rate for a sufficiently long time, the distribution over final model weights will converge towards a set of delta-functions at the loss function minima (~ a low temperature Boltzmann distribution, under some assumptions on the batching noise). In this case, when the loss is perturbed, the best possible transport map (in terms of KL-divergence) will simply translate these delta functions to their new locations. Thm 3 shows that this best map is given by influence functions. Thm 2 shows that unrolled differentiation converges to the same IF formula. Hence, when training for sufficiently long we get that ‘optimal transport map’ ≈ ‘influence functions’ ≈ ‘unrolled differentiation’. We will add this discussion to the text; thanks for prompting this.
>
> **Citations**. Thanks for the extra references, which we have now added.
>
> Once again, we warmly thank the reviewer for their comments. If the reviewer thinks the newly added experiments and explanations strengthen the paper, we respectfully ask that they consider raising their score.

---

> > ### Comment · Reviewer_A7Ts · 2025-08-04
> >
> > I thank the authors for their comprehensive rebuttal.
> >
> > The newly added experiments on Vision Transformers and diffusion models strengthen the paper.
> >
> > The clarifications on the theoretical assumptions and the relationship between Theorems 2 and 3 are also well-received.
> >
> > I would like to maintain my positive score.

---

### Official Review · Reviewer_sTvu · 2025-07-01

**Clarity:** 3
**Significance:** 2
**Originality:** 2
**Rating:** 4
**Confidence:** 3

**Summary:**

This paper introduces the idea of distributional data influence (d-TDA), which characterizes the potential distribution of the result model when removing some training samples from a stochastic training algorithm. The authors present an efficient d-TDA method based on unrolled differentiation and evaluate its ability in estimating the ground truth distribution change. Finally, the paper presents a lot of empirical analysis of the estimation results and provide a rigorous analysis on how the well-known influence function naturally arose in this setting.

**Questions:**

See weakness and some more questions:

1. In line 150, typically how many trained models are needed to accurate estimate the overall distribution? I would like to clearly understand the estimator's cost.
2. The current unrolled differentiation framework assumes SGD training, will the faithfulness of the result be severely damaged if trained with other optimizers such as AdamW or Muon?

This is a very interesting work, but my main concern still revolves around the practical usability of this framework comparing to the traditional non-distributional ones. If the author could add some experiments or comparisons to better illustrate this point I am happy to raise my score.

**Ethical Concerns:**

["NO or VERY MINOR ethics concerns only"]

**Final Justification:**

The paper introduces a novel data-influence framework called distributional data influence. The work is overall technically sound and rigorous. The evaluation can successfully demonstrate the advantage of d-TDA over traditional TDA with the promised experiments on ViT, though they are still not very comprehensive. The presentation of the paper can also be further improved, as I see many reviewers share the same confusion which is not well-answered in the current draft. Overall I think the merit of this paper is bigger than its limitations, and I thus give a score of 4.

**Limitations:**

yes

**Quality:**

3

**Strengths And Weaknesses:**

**Strengths:**
1. The paper generalizes training-data attribution from single-model effects to shifts in the entire distribution of models induced by training randomness, which is a natural and interesting setting.
2. The exposition is clear and technically rigorous, with some insightful theoretical analysis.

**Weakness:**
1. Lack of numerical results: Most experiments are conducted only on toy regression tasks, and the biggest dataset/model tested is on the MNIST dataset. This raises question on how practical is the current methodology to the current large machine learning models.
2. The paper currently does not make any experimental comparisons of the distributional method to standard (non-distributional) attribution (perhaps some downstream tasks such as dataset pruning or noisy-label detection?), so the claimed advantage remains speculative. If the new distributional setting offers little improvement while significantly increasing computational cost, its practical value remains doubtful despite its theoretical appeal.

---

> ### Author Rebuttal · Authors · 2025-07-31
>
> We thank the reviewer for their thoughtful feedback, and are pleased that they identify the work as interesting, clear and technically rigorous. We address all their comments and questions in detail below.
>
> **Few numerical results.** The reviewer is right to note that our chief contributions are conceptual and theoretical, offering some preliminary empirical evidence via UCI concrete and MNIST but deferring more exhaustive experiments to future work. _Our main intention is to introduce and explore the mathematical foundations of d-TDA – especially, as a lens to understand the unreasonable effectiveness of IFs in deep learning_. Given how widely influence functions are being used in deep learning, and the limitations of current interpretations, we believe this is a major theoretical contribution. Future papers can further explore applications and improvements to IFs based on insights and evaluations suggested by framework.
>
> That said, we do agree that the paper could benefit from some additional empirical results. For this reason, following the reviewer’s comments, we have added **additional experiments applying our d-TDA methods to transformers and diffusion models**, including the suggested **data pruning** task with vision transformers. For the vision transformer experiments, we show that distributional (mean) influence can better identify training datapoints that will improve the final test loss _fo a single model_ when removed from the training set. By removing 10% of datapoints, we can improve the test loss from 0.715 to 0.509 (compare to 0.528 for regular TDA with a fixed seed). For latent diffusion trained from scratch on a dataset of 50000 Artbench images (256x256 resolution), we compare top predicted Wasserstein shift and mean shift samples, illustrating that they affect the ground-truth retraining distribution in the predicted manner.
>
> **Comparison to standard (non-distributional) TDA**. Thanks for the comments. We again emphasise that our chief goals are to 1) introduce a new training data attribution framework that properly takes into account stochasticity, and 2) show how existing standard methods such as IFs can be properly motivated and used within this mathematically rigorous framework, in contrast to the non-distributional TDA setting.  Whilst we do spend some time comparing distributional influence to its regular counterpart (see especially Fig 7 in the appendix, where ‘mean influence’, albeit distributional, is the current standard practice), our principal focus is not new data attribution methods per se, but rather a novel way of thinking about existing and future training data attribution methods.
>
> That said, we agree showing the practical limitations of applying IFs to a single model with a fixed random seed would be beneficial. We believe **the newly added data pruning experiments with vision transformers on CIFAR-10**, showing empirical improvements from using distributional influence, illustrate this point.
>
> **Questions**. Thanks for the great questions.
>
> *How many samples are needed for d-TDA?* One can adopt a distributional perspective with only 1 sample. In this case, the original sample is interpreted as distributed according to the measure for the original dataset. Meanwhile, the modified sample, obtained by offsetting with IFs or unrolled differentiation, is (approximately) distributed according to the measure for the new dataset. Through Thms 2 and 3, we make this statement mathematically rigorous for the first time. More pragmatically, if computing e.g. Wasserstein or mean influence, we typically use 5-100 samples, with full details for each experiment reported in the Appendix. Also note that practitioners often heuristically average over a small ensemble of IFs to improve performance. Hence, one could argue that we are really just suggesting a more rigorous way to use this existing set of samples, at essentially identical computational cost.
>
> *Does d-TDA work with other optimisers?* Yes. Unrolled differentiation can be extended to arbitrary optimisers, provided their updates are differentiable. We believe our Thm 2 can be extended in kind. In fact, following the reviewer’s comments, we have verified that this is the case for SGD with momentum. However, we leave a full exploration of the theoretical details to important future work, following standard practice in the literature by initially focussing on SGD (Bae et al, 2024).
>
> We again thank the reviewer for their time and comments. With the above clarifications in mind, as well as the extra experiments and theory added in response to their comments, we respectfully ask that they consider raising their score and recommending acceptance.
>
> **References**.
> (Bae et al., 2024) Training Data Attribution via Approximate Unrolled Differentiation, https://arxiv.org/abs/2405.12186

---

> > ### Comment · Reviewer_sTvu · 2025-08-02
> > **Official Comment by Reviewer sTvu**
> >
> > I appreciate the author for their detailed responses. My previous main concerns echoed those of reviewer wFZD: it was unclear whether d-TDA truly improves on conventional fixed-seed TDA, which is why I requested additional comparisons. I think that through the additional experiments with vision transformers on CIFAR-10, the authors address this concern well. Therefore, I will raise my score to 4.
> >
> > However, I wish to say that the presentation of this paper could be improved, as I feel like I share a lot of confusion with other reviewers. For example, I have the same impression as reviewer gp8r that d-TDA is much more costly than normal IF, and I was similarly confused about the position of IF in this paper as reviewer wFZD. But overall I feel positive of this work, and believe it is technically novel and rigorous.

---

> > > ### Author Response · Authors · 2025-08-07
> > >
> > > Many thanks for your reply! We are pleased that our extra experiments have addressed your concerns about whether d-TDA improves conventional fixed-seed TDA, and that you feel positive about the work overall.
> > >
> > > > [...] the presentation of this paper could be improved [...] I was similarly confused about the position of IF in this paper as reviewer wFZD. But overall I feel positive of this work [...]
> > >
> > > Thanks for raising this remaining concern. We strongly agree that the messaging needs to be clear -- it is difficult to fit so many complex mathematical ideas into such a short space! Thanks for bearing with us. In response to wFZD, we have reframed the introduction to emphasise the following:
> > >
> > > 1. IFs are already secretly distributional, which helps explain why they work in deep learning (where training is stochastic and losses are non-convex)
> > > 2. _Explicit_ distributional TDA is also independently interesting, and can bring benefits in certain applications (ViTs, diffusion, etc.)
> > >
> > > Concretely, the new 'core contributions' paragraph now reads as follows:
> > >
> > > >Core contributions: 1) We introduce _distributional training data attribution_ (d-TDA), a framework for studying data attribution in stochastic deep learning settings. We show that d-TDA helps explain the limitations of leave-one-out retraining, and the effectiveness of influence functions (IFs). 2) In particular, we prove that IFs actually solve certain distributional TDA tasks, approximating a) the limiting distribution of unrolled differentiation and b) the best possible transport map between the final (Boltzmann) distributions of model weights, under select assumptions. 3) Lastly, within the d-TDA framework, we propose _distributional influence_, which quantifies the importance of training examples by how much their inclusion/exclusion affects the distribution over model weights and outputs (Section 3). We show that distributional influence can be used to benchmark TDA methods and that it captures interesting information missing from its regular predecessor. We provide preliminary empirical evidence that our novel distributional influence variants are effective in data pruning tasks.
> > >
> > > We hope you agree with us (and reviewer wFZD – please see replies therein!) that these updates improve the presentation, and very much hope that such refactoring resolves any remaining confusion and concerns. If it does, we kindly ask that you consider a further score increase. Otherwise, please do let us know how we could further improve the paper or be clearer on the above points!

---

> > > > ### Comment · Reviewer_sTvu · 2025-08-09
> > > >
> > > > Thanks for the thoughtful updates. This version indeed reads much clearer.

---

### Official Review · Reviewer_wFZD · 2025-07-02

**Clarity:** 2
**Significance:** 2
**Originality:** 3
**Rating:** 5
**Confidence:** 3

**Summary:**

This paper focuses on the problem of training data attribution (TDA), often also called "influence estimation." The goal of TDA is to estimate the impact of removing a training datapoint on the model parameters or some function of the model parameters. The paper notes that previous methods have ignored or glossed over the randomness present in training modern machine learning models via the random initialization and batch ordering. In particular, existing TDA methods focus on providing a point estimate that estimates the influence for a datapoint given a particular random seed. The paper proposes to instead compute *distributional* TDA (d-TDA or distributional influence), which computes the influence under all possible random seeds. In synthetic experiments, the paper shows that distributional influence can provide different answers than does regular influence. The paper then proves that influence functions (the typical method for estimating regular influence) are in fact a limit of distributional influence in that they approximate the d-TDA distribution.

**Questions:**

My main questions are:
1. When is it practically meaningful to compute distributional influence?
2. What is the paper's position on the use of IFs?
3. How is the paper proposing to compute IFs? If the answer is the algorithm on lines 150-155, what exactly is going on in this algorithm?

Thank you!

**Ethical Concerns:**

["NO or VERY MINOR ethics concerns only"]

**Final Justification:**

The authors have, I think, addressed a lot of the issues brought up in the reviews. I think the re-framing of their work, as well as the new experiment showing the potential for d-TDA to provide practically meaningful guidance have really made the paper's contribution a lot more clear. So, I've updated my score to vote for an accept.

**Limitations:**

Yes

**Quality:**

3

**Strengths And Weaknesses:**

I found this to be a technically sound paper that computes an interesting quantity: the distributional influence of a datapoint. As far as I'm aware, this is a novel contribution that hasn't been considered before. And the paper successfully demonstrates an example where the use of distributional influence results in different estimates than the use of regular influence, which is an important step for this type of algorithm being practically useful.

**Overall motivation**

One of the biggest issues I see with the paper is that it's unclear why one would want to compute distributional influence. As far as I'm aware, one is generally interested in influence of a datapoint over the finished end model so that, e.g., one can wonder about whether a datapoint is negatively contributing to its performance (for the purposes of re-running the training) or if a datapoint was valuable to its performance (for the purposes of compensating data providers). I don't see a clear argument for anyone being interested in knowing how valuable a datapoint was for a hypothetical model with different number of layers, order of layers, activation functions, number of training epochs, or other hyperparameters. And I see random seed as just another hyperparameter that defines the particular model one gets after training. So why is it of interest to get results under some distribution of this hyperparameter? I don't think the paper currently makes this case, but instead points out that distributional influence is different, and then concludes that this means that it is meaningful. For example:

- Figure 3 shows that regular influence functions capture different information than d-TDA. The paper concludes that "This showcases that d-TDA captures meaningful information about the training data missed by previous notions of influence." I agree Fig 3 shows it captures different information, but I don't see why Fig 3 shows this information is *meaningful* for any tasks.
- "The ability to predict how different training examples impact the distribution over training runs may be practically useful. For example, one could identify examples to remove to reduce variance, improving model repeatability over training runs." Why is this practically meaningful? At first thought, variance over training runs sounds like a good thing because it allows us to find more local optima, some of which may be better than our initial training run. So why should a practitioner care if they can consistently get to the same local optimum using different random seeds? They already know how to get to that local optimum.

I also have two other points about the motivation of the paper:

- I'm not sure exactly what the takeaways from Theorem 3 are. I would say that Thm 3 shows that, for Boltzmann distributions, IFs are optimal to use. But why do I care about Boltzmann distributions? How does this relate to a user thinking of using IFs for some model in practice?
- I can't tell what the paper's position on IFs is. Earlier in the paper, it seems to be that they're not good because they're not measuring distirbutional influence. But then, in the final paragraph before the conclusion, the paper seems to turn around and say that the whole paper motivates the use of IFs. So is the paper providing anything besides suggesting the use of IFs? Maybe its algorithmic contribution is to use unrolled differentiation, but I believe previous work has covered this (see Luo et al. (2023)).

**Related work**

The work of Luo et al. (2023) is highly related to this paper and its discussion of unrolled differentiation. Luo et al. (2023) study the approximation of LOO in models fit with SGD, and essentially use similar unrolled differentiation techniques as proposed in Eq (6). So I think the quote from line 164 of the current paper, "To the best of our knowledge, this is the first application of such techniques" should be updated.

The paper notes that previous works like [20] argued that IFs are not good for predicting leave-one-out retraining for deep learning models. The paper claims that this is incorrect, and LOO is in fact just noisy, rather than IFs being bad. Is the claim here that d-TDA applied to the situations in [20] would work well? I think this is a major empirical claim that would be pretty impactful if true. But big claims need big evidence, and there's no discussion or empirical evidence backing this up; it feels like investigating this claim could be an entire paper on its own. I think this paragraph either needs to be removed or needs significant expansion.


**Unclear things**

I was unclear about what some of the experiments were trying to show, as well as found a few other places where things were unclear; I've detailed the issues below:

- The algorithm proposed to compute d-TDA is on lines 150-155. I'm not clear on what the algorithm is. The key Step 3 says "using these two sets of (correlated) samples...", but what are the two sets? It seems like only step 2 describes a set of samples.
- Figure 2 is supposed to be an initial empirical explanation of d-TDA, but I wasn't clear what Figure 2 is displaying. First, what is "query sample" in the legend? Are the models one-dimensional here? Because it seems like the leftmost figure is supposed to be plotting models. Next, the caption states that the middle figure shows that the exact retraining samples are closer to predicted samples than samples from the original model. But the center figure seems to only be showing two datapoints per color here. To show thing $a$ is closer to thing $b$ than is thing $c$, I would think we need three points on the plot ($a,b,c$) for each color. Finally, on the right, I'm not sure what each of the terms in the legend means.
- Figure 4 also uses the term "query point," and I'm not sure what it means here either. I tried searching the text for the word "query", but it's only used once: in the caption of Fig 2.
- Figure 5 (top) claims to show measurements predicted by unrolled differentiation versus changes predicted by IFs. But I only see one set of lines here. Are the dashed lines supposed to be one unrolled differentiation or IFs? But then why do only some of the lines have corresponding dashed lines? Also why is the top line so thick?  Finally, I see only three lines, but there are five lines listed in the legend.
- Figure 5 bottom looks like it has a bunch of things overlapping. Are these supposed to correspond to the different learning rates?
- Sometimes $k$ seems to index datapoints and sometimes $i$ seems to index datapoints.


**Theoretical development**

I thought there were a few small issues with the theoretical development:

- Assumption A1 says that the loss function gradients $\nabla \ell_k$ are "Lipschitz continuous and bounded". Bounded for each $k$? Or uniformly? Also, is it true for neural networks that their Hessian is bounded and Lipschitz?
- "If the [model] weights converge to a saddle point ... the final parameters will become sensitive to any infinitesimal perturbation of the loss function." I'm pretty sure this isn't true. Consider the loss $L(\theta) = \sum_{n=1}^N (x_n - \theta)^3$. If $x_1 = \dots = x_N = 0$, then there is a saddle point at $\theta = 0$, and this is still a saddle point even after you remove any given $x_n$.
- I didn't see any motivation for assumptions A4 and A5. Should we expect these to hold up in cases where we're typically using SGD? I see the discussion in 291-298 describes why they are useful for the proof, but why are they reasonable?
- Duplicate assumption numbers are used -- there are two A1, A2, A3, A4, and A5's in the paper.
- The second A2 states "the nonzero eigenvalues of the Hessian are uniformly bounded away from zero." How could this be violated? The Hessian is of finite dimension, so the minimum non-zero eigenvalue is bigger than zero.
- Theorem 3 uses the term "minimum manifold," but I don't think this was defined.


**References**

Yuetian Luo, Zhimei Ren, and Rina Barber. Iterative Approximate Cross-Validation. ICML. 2023

---

> ### Author Rebuttal · Authors · 2025-07-31
>
> We thank the reviewer for thoughtful comments and for finding our work technically sound and novel. We address their main concerns below and highlight our **new diffusion and transformer experiments**, added in response to their feedback.
> # Motivation for d-TDA
> **Why is distributional influence interesting?** It is widely acknowledged that conventional TDA methods are limited by their inability to account for stochasticity in training (K et al., 2021, _Revisiting Methods for Finding Influential Examples_). It is difficult to distinguish changes to model behaviour due to training data modifications from changes due to sampling randomness. Furthermore, usual derivations of IFs do not account for non-convex losses with multiple minima (Bae et al., 2022, _If Influence Functions are the Answer, then What is the Question?_) which are typical in deep learning. In contrast, in the d-TDA setting IFs can be naturally derived without these restrictive assumptions.
>
> **Why not regular TDA with a fixed seed?** This is an important question that we will dedicate a discussion to in the paper. Even with a fixed seed, chaotic training dynamics can push training to a different parameter space region when perturbing a dataset. To aggrevate this, in standard deep learning training with a fixed batch-size, training on a subset completely offsets at what iteration each datum is presented, even with a fixed seed. There are no guarantees that in these settings IFs would give us an answer to the counterfactual retraining question with a fixed seed. This is why prior work relies on convexity assumptions, where everything converges to the same point. In contrast, our work rigorously shows that IFs do something meaningful _even in this challenging setting_. We just have to interpret IFs as approximate resampling instead.
>
> To illustrate this, we will add an experiment where we vary the peak learning rate at the beginning of training, showing the break-down in correlation between IF prediction and retraining _with the same seed_. Nonetheless, we illustrate that IFs are good at approximately resampling from the target distribution in all cases. Furthermore, *our new vision transformer data pruning experiments* illustrate that distributional (mean) influence does better in practice than applying IFs to a single model with a fixed seed.
>
> We also think that the distributional properties of training may be of *independent interest*. Practitioners might be interested in selecting the data to reduce the variance for the output for a given unlabelled input, or they might want to do data selection to ensure it leads to improvements in validation loss for 90% of training runs. Our d-TDA methods allow practitioners to curate the dataset to target both these desiderata, or yet more complicated distributional properties. **We demonstrate this with new Latent Diffusion and vision transformer experiments on 50000 Artbench (256x256 resolution) and CIFAR-10 datasets respectively.**
>
> We agree that exploring the full scope d-TDA applications is an important future research direction. However, our paper's chief goal is to build the mathematical foundation of influence functions for deep learning through the lens of d-TDA.
> # Takeaways from theorems and position on IFs
> **Purpose of Thm 3.** Thm 3 shows that IFs are the optimal KL-divergence approximation to the d-TDA question if the fully trained model weights obey a Boltzmann distribution with the energy given by the loss – a reasonable approximation which arises from approximating the batching noise in SGD (Mandt et al., 2017, _Stochastic Gradient Descent as Approximate Bayesian Inference_). In this setting, IFs provide precisely the KL-divergence-optimal asymptotic transformation to map samples trained with the original dataset to the samples trained with the perturbed dataset. The isotropic Gaussian noise is a simplifying relaxation, akin to how prior works have studied convex losses (albeit arguably a weaker one). We include Thm 3 to demonstrate that IFs have another distributional motivation (similarly to Thm 2). By slightly simplyfing the setting compared to Thm 2, we obtain a stronger result: IFs are the optimal distributional approximation in terms of KL-divergence. It is not intended to be interpreted as practical guidance for training models.
>
> **Position on IFs (Q2).** The core motivation for this work is that IFs appear to work well in downstream tasks, but lack theoretical and conceptual motivation in deep learning due to stochastic training and non-convex losses. Our paper provides a justification for IFs in these settings by showing that they can be interpreted as _resampling_. Hence, our paper is pro-IFs in that *it provides new strong motivation for their use through links to unrolled differentiation*.
>
> ## Minor points
> _Related work of Luo et al. (2023)._ Forward-mode unrolled differentiation in deep learning was proposed even earlier by (Franceschi et al., 2017, _Forward and reverse gradient-based hyperparameter optimization_), whom we cite. We claim that our paper is the first such application _to TDA_ (hence ‘first application [...] for the computation of the response’). We have clarified this; thanks.
>
> _Empirical evidence that LOO is just noisy._ Please see Fig. 6 and App. C.1, in which provide the suggested experiments.
>
> _Figures 2, 4 and 5_. We apologise that parts of the figures were unclear to the reviewer. The captions are dense due to the intial 9-page limit.
>
> _Fig 2_. We consider neural networks trained on UCI concrete, with either the full dataset or a subset. We consider 4 measurements: the 1D neural network output on 4 chosen ‘query samples’. Each is represented by a different colour. **Left:** we show actual outputs trained on a subset (x axis) plotted against against predicted outputs (y axis) for each measurement (colour). The prediction for the outputs trained on a subset (“retrained”) are made by applying unrolled differentiation to models trained on the full dataset. Each sample corresponds to a single random seed. Here, a perfect prediction would lie on the black $x=y$ line – meaning predicted matches ground-truth retraining. Most samples lie far away from the line. **Centre:** Repeats the left panel, but shows the mean over all random seeds. We see that the _mean of each measurement distribution predicted with unrolled differentiation_ (circles) actually lies close to the $x=y$ line, certainly closer than the mean of the original measurement distribution of a model trained on all the data (diamonds). **Right:** we show that the true retrained distribution (colour, line) is closer to the predicted retrained distribution (colour, shaded) than the distribution trained on the full dataset (black). We will make sure this is explained more thoroughly in the caption.
>
> _Fig 4_. As in prior work, ‘query point’ refers to an example (input) for which we compute the measurement function. We now introduce the term explicitly.
>
> _Fig 5_. As the y-label indicates, the top plot shows the Pearson correlation between IF-predicted and UD-predicted measurement changes (y-axis) against training time (x-axis). 5 differently-colored lines correspond to runs with different learning rates. Three smallest learning rates give similar correlations, so they overlap. We have made this visually clearer. The dashed lines are spurious and have been removed.
>
> > Also, is it true for neural networks that their Hessian is bounded and Lipschitz?
>
> This is a common assumption in deep learning theory (_far_ weaker than convexity). For instance, it holds if we consider optimisation on a bounded domain (which is virtually always true in practice due to finite-precision arithmetic).
>
> > "If the [model] weights converge to a saddle point ... the final parameters will become sensitive to any infinitesimal perturbation of the loss function." I'm pretty sure this isn't true.
>
> We mean that a gradient flow trajectory converging to a saddle point will *in general* (not always) be unstable under infinitesimal perturbations to the loss function. One can construct careful perturbations for which this is not the case. To illustrate, consider $L(\theta) = \theta_1^2$. The set $\{\theta \in \mathbb{R}^2: \theta_1=0\}$ are all minima, but any infinitesimal perturbation $L(\theta) + \epsilon \theta_2^2$ collapses the minimum set to a single point. We have now phrased this more carefully in the text.
>
> _Motivation for assumptions A4 and A5_. A4 is intuitive as it's necessary for unrolled differentiation to not diverge; if unrolled differentiation diverges then we certainly cannot expect it converges to IFs. Likewise, A5 is needed to make sure that minima cannot be arbitrarily flat so IFs themselves do not diverge. We stress A1-A5 are intended to describe sufficient conditions for unrolled differentiation to converge to IFs. Whilst reasonable for many deep learning applications, we certainly do not claim that they will hold universally. Indeed, situations where they do not hold may be of independent interest for understanding when IFs break down. We'll make sure to expand the discussion of the motivation of each assumption.
>
> _Bounding nonzero Hessian eigenvalues away from zero._ A5 refers to the nonzero Hessian eigenvalues on the set of critical points (‘minimum manifold’), rather than at a single location $\theta$ – hence why the assumption is nontrivial.
>
> _Missing definition._ The set $\mathcal{S}^g_\mathcal{L}$, to which ‘minimum manifold’ refers, is defined in line 339; we made the connection clearer.
>
> We thank you for your thorough feedback. If anything remains unclear, please let us know. We hope our clarifications and new experiments warrant recommending acceptance.

---

> ### Author Response · Authors · 2025-08-05
>
> Thanks for the detailed response and for continuing to engage so wholeheartedly with the reviewing process! We completely understand your confusion, and agree that the text needs refining to be clearer on these points.
>
> To answer your question: the paper intends to do both of (1) and (2). In our minds, the core contribution of the paper is that IFs appear naturally if you try to tackle data attribution in a rigorous, distributional manner for general, non-convex loss functions -- or, as you aptly put it, they are 'secretly an approximation to d-TDA' (1). Of course, properly introducing d-TDA (2) is a prerequisite for this, which we also think may be of independent interest. Taking the distributional interpretation of IFs to its logical conclusion, a secondary contribution is to come up with _distributional influence_: metrics to measure how effective TDA methods are at estimating changes in distributions over model outputs (2). Distributional influence can be used to benchmark how effective methods like IFs and unrolled differentiation (or others) are at this job.
>
> As a minor point, which may be helpful for understanding our contributions: we emphasise that we use ‘distributional _TDA_’ -- a framework for thinking about TDA with stochastic training algorithms -- distinctly from ‘distributional _influence_’ -- a measure of how important individual training examples are in a distributional sense.
>
> We agree that (1) should be the main focus of the paper. (1) is where we dedicate most of our mathematical work. However, we think that (2) is also a notable contribution. A proper distributional TDA framework is essential to even understand contribution (1), and moreover our extra experiments show that distributional influence can be practically useful. For a dataset pruning task with a ViT trained on CIFAR-10, **distributional (mean) influence can reduce the test loss from 0.725 to 0.509 — compared to 0.528 with non-distributional ‘fixed-seed’ variant** — for a single model trained with the same seed. In this sense, distributional influence is very much a 'new, real algorithm' that can be practically useful -- even if we do not necessarily recommend that it should be used in every possible instance (e.g. LLMs).
>
> As a subtle but important point: we emphasise that (2) should _not_ be interpreted as a weakness of IFs per se, because one can actually use IFs to compute distributional influence. As we say in the text (lines 361-366), you can approximate samples from the retrained model by taking samples from the original model and offsetting each of them by $r_{IF}/N$. Regular IFs are thus fully compatible with d-TDA, and with computing distributional influence. It is not a case of 'IFs vs d-TDA', but rather 'IFs for d-TDA'. When we talk about 'classical IF/TDA' methods, we mean using IFs naively without following the algorithm in lines 150-156. Indeed, even the regular usage of IFs with just one seed can actually be seen as a special case of distributional influence: namely, unbiased estimate of mean influence with a single sample. This may help explain why they work in practice for non-convex losses.
>
> With the above discussion in mind, we would like to suggest renaming the paper to:
> - Why Do Influence Functions Work? A Distributional Perspective
> - Influence Functions as Distributional Data Attribution
> - The Distributional Data Attribution Task and Why Influence Functions already Solve It
> or similar, subject to discussion with reviewers and other authors.
>
> We will refine the text to more clearly convey what our contributions are. As an illustrative example, we have sharpened the 'Core Contribution' paragraph (L53-L62) as follows:
>
> > Core contributions: 1) We introduce _distributional training data attribution_ (d-TDA), a framework for studying data attribution in stochastic deep learning settings. We show that d-TDA helps explain the limitations of leave-one-out retraining, and the effectiveness of influence functions (IFs). 2) In particular, we prove that IFs actually solve certain distributional TDA tasks, approximating a) the limiting distribution of unrolled differentiation and b) the best possible transport map between the final (Boltzmann) distributions of model weights, under select assumptions. 3) Lastly, within the d-TDA framework, we propose _distributional influence_, which quantifies the importance of training examples by how much their inclusion/exclusion affects the distribution over model weights and outputs (Section 3). We show that distributional influence can be used to benchmark TDA methods and that it captures interesting information missing from its regular predecessor. We provide preliminary empirical evidence that our novel distributional influence variants are effective in data pruning tasks.
>
> We again thank the reviewer for their thoughtful feedback – it has helped improve the clarity of the writing!

---

> ### Comment · Reviewer_wFZD · 2025-08-07
>
> Thanks for continuing to respond in so much detail!
>
> This rephrasing makes a lot more sense to me. I think presenting the paper along the lines proposed here really alleviate a lot of my issues, and I've increased my score to vote for an accept.
>
> The only point above that I think could use more support in the paper is"We show that d-TDA helps explain the limitations of leave-one-out retraining" (either more experiments / deeper discussion of literature / logical arguments). But it's not totally unsupported by the paper, so I don't think this is a big issue. And overall I think the flow of "surprise! IFs are distributional, and by the way, doing explicitly distributional TDA might be of separate interest" is a great contribution to this literature.
>
> (Edit: I don't have any complaints about the current title of the paper, so I'd encourage the authors to update it only if they want to!)

---

### Official Review · Reviewer_gp8r · 2025-07-02

**Clarity:** 3
**Significance:** 3
**Originality:** 4
**Rating:** 4
**Confidence:** 4

**Summary:**

The paper introduces Distributional Training Data Attribution (d-TDA), a new framework for attributing model behavior to training examples under stochastic training dynamics. Unlike classical methods like influence functions that assume deterministic training and focus on mean effects, d-TDA models the full distribution over model outputs induced by random initialization and batch sampling. The authors define distributional influence as the change in output distribution (e.g., mean, variance, Wasserstein distance) when a training point is removed. They propose an efficient approximation using unrolled differentiation, avoiding retraining by computing a first-order response that estimates distributional shifts. The paper also provides new mathematical justifications for influence functions, showing they arise as limiting cases of d-TDA under certain conditions. Empirical results demonstrate that d-TDA captures richer signals—such as uncertainty changes—that traditional methods miss. This work redefines data attribution by treating randomness as a feature, not a nuisance, and opens up scalable and principled analyses for deep learning models.

**Questions:**

Q1: Do you think the actual distribution of influence function is achievable in practice?

Q2: Whether you think the estimation of $\theta^*(D\z_k)$ is always necessary, since we sometimes just need to estimate the training samples contribution to a specific prediction on a testing sample. For example, I am aware that paper which bypass such estimation for empirical convenience, which can locate the real problems in public data sets.

> Debugging and Explaining Metric Learning Approaches: An Influence Function Based Perspective (NeurIPS 2022)

In their approach, they just find a $\theta$ where a training sample and a testing sample are mostly likely to be correlated.

Q3: I like your interpretation on the leave-one-out training, but what is the useful measurement for real-world experiment of data attribution?

Q4: In practice, how your d-TDA can be significantly different from classical TDA to identify problematic training data samples?

**Ethical Concerns:**

["NO or VERY MINOR ethics concerns only"]

**Final Justification:**

I am still supportive for this work, which is interesting and insightful.

**Limitations:**

It is overall a wonderful paper with solid theoretical contribution and insights. However, it seems to be limited in answering how this work can empirically support real-world application.

**Quality:**

4

**Strengths And Weaknesses:**

# Strengths and Weakness
Strength:
+ highly theoretically interesting
+ theoretically sound
+ new insights into many aspects of influence function


Weakness
- the IF calculation to become even more computationally expensive
- it is not clear how could we sample seeds for *actual* distribution, which leads to concern whether the ground-truth distribution is achievable
- little evidence to show the distribution is really necessary for data attribution in practice

---

> ### Author Rebuttal · Authors · 2025-07-31
>
> We thank the reviewer for their favourable comments on the text. We appreciate they describe our perspective on IFs and data attribution in deep learning as “highly theoretically interesting” and “theoretically sound” -- key goals of the paper. We are happy to address their concerns and some points of minor misunderstanding below, also emphasising the **extra transformer and diffusion experiments** added in response to their helpful feedback.
>
> > 1. the IF calculation to become even more computationally expensive
>
> We stress that a core contribution of the paper is showing how standard influence functions (IFs) can already be interpreted as resampling within a distributional framework (Thms 2 and 3). Concretely, offsetting trained model weights by an IF enables one to (approximately) sample from the distribution of models trained with a modified dataset. Our paper makes this claim mathematically rigorous. Hence, our work shows that standard influence functions applied to a single model are *already distributional*. In this sense, d-TDA is not inherently more expensive than IFs: d-TDA simply provides a new interpretation of what IFs actually do in deep learning. Without this distributional interpretation, IFs must be derived with unrealistic assumptions on the loss function: e.g. convexity or the presence of a unique minimum (e.g. the PBRF approximation in (Bae et al., 2022)).
>
> The distributional reinterpretation allows us to *in addition* define novel measures of distributional influence. In our minds, the primary utility of of distributional influence is for *evaluating* methods like IFs in deep learning. Since IFs are inherently distributional, we think distributional evaluations are appropriate to verify how well they approximate actual resampling. As the reviewer correctly notes, calculating e.g. Wasserstein influence does indeed require an ensemble of samples – as few as 5 suffices in some of our experiments. However, when evaluating TDA methods, it is already standard practice to average TDA results over a small number of models (Park et al., 2023). Hence, our algorithm does not actually substantially increase the cost compared to usual in these cases.
>
> > 2. It is not clear how could we sample seeds for actual distribution, which leads to concern whether the ground-truth distribution is achievable
>
> Could the reviewer kindly clarify what they mean by “how could we sample seeds for actual distribution”? If you are asking how we sample from the “ground-truth” training distribution trained on a given dataset: to obtain a sample, we change the seeds for the training sources of randomness – 1) the data batch ordering, 2) model initialisation, and 3) any other sources of stochasticity (e.g. dropout masks). This is what we report as grount-truth samples in e.g. Fig. 2.
>
> > 3. ‘Little evidence to show the distribution is really necessary for data attribution in practice.’
>
> The reviewer is right to note that this initial work focuses on introducing the conceptual foundations of d-TDA and understanding its mathematical properties. Whilst we provide some empirical demonstrations, we do indeed defer a full–scale experimental evaluation of downstream applications as important future work. We again stress that the crux of our paper is about showing that influence functions inherently answer a distributional question; we do not argue that everyone should always estimate the whole distribution for every downstream task.
>
> That said, we agree that the paper might benefit from a clearer demonstration of the downstream utility of distributional influence. Therefore, in response to the reviewer's feedback, **we added an extra dataset pruning experiment comparing distributional influence against regular influence on a subset selection task with SWIN Vision Transformers on CIFAR-10**. Our results show that, by using distributional (mean) influence, we can select a better subset of examples to reduce the test loss than when we compute influence on a single model with a fixed seed. Our distributional subset selection  removing 10% of data improves the final test loss from 0.725 to 0.509 (compared to 0.528 with non-distributional variant). We use 10 samples to estimate the distribution.
>
> ## Questions:
> >Do you think the actual distribution of influence function is achievable in practice?
>
> Yes. Our novel theoretical results show that you can approximately sample from the distribution over retrained models with as few as 1 training run and its corresponding IF (see Thm 2). Moreover, taking a small ensemble of models (~5 models), one can effectively capture changes to the distribution over model measurements (Fig 2 right). We again stress that training with multiple seeds is the default practice when evaluating methods like IFs (see Appendix C.2).
>
> > Whether you think the estimation of [influence on parameters] is always necessary, since we sometimes just need to estimate the training samples contribution to a specific prediction on a testing sample.
>
> Thanks for the comment. We certainly agree that there is an important place in the literature for empirical debugging and explanation tools, like those referenced by the reviewer. Such techniques are sometimes framed as analogous to IFs when used for TDA. However, our paper is more concerned with the following mathematical question: how does the distribution over final model predictions change depending on the dataset? Whilst we agree that fast, heuristic alternatives to unrolled differentiation and IFs might suffice in certain down-stream applications, these techniques do not necessarily address this question. There are many cases where we really care about understanding how the model would behave if retrained, e.g. copyright concerns (Mlodozeniec et al., 2024) or understanding LLMs (Grosse et al., 2023). As such, we need to develop principled methods that aim to answer this question. In this sense, we consider these works to be orthogonal to our work.
>
> > I like your interpretation on the leave-one-out training, but what is the useful measurement for real-world experiment of data attribution?
>
> We are pleased the reviewer appreciates our contribution that LOO is not inherently incompatible with IFs:  a distributional interpretation reveals that the problem is just a low signal to noise ratio (SNR), and with enough samples IFs can indeed approximate the LOO distribution. For evaluation purposes, we believe leave-many-out retraining (as is standard practice in the literature (Park et al. 2023)) is a useful alternative for real-world experiments that has a sufficient SNR. We report such experiments in Appendix C.2.
>
> > In practice, how your d-TDA can be significantly different from classical TDA to identify problematic training data samples?
>
> Fig 3 already demonstrates an example of a datapoint whose exclusion does not substantially modify the mean model behaviour, but which does drastically modify the distribution over model outputs in e.g. the Wasserstein sense. We provide further examples for MNIST in Fig 4. Training examples that are visually and intuitively ‘influential’ are sometimes missed by mean TDA. **In response to this comment, we have now also added experiments comparing top Wasserstein shift vs mean shift samples on Latent Diffusion Models on 50000 Artbench images (256x256 resolution)**. Once again, we find that most important examples do differ between Wasserstein and mean influence.
>
>
> ---
>
> We again warmly thank the reviewer for their feedback. We hope we have resolved all their questions and concerns, but invite them to let us know if any remain. If satisfied with our clarifications and the extra experiments, we respectfully ask them to consider raising their score.
>
> ## References
> (Park et al., 2023) TRAK: Attributing Model Behavior at Scale, https://arxiv.org/abs/2303.14186
> (Bae et al., 2022) If influence functions are the answer, then what is the question?, https://arxiv.org/abs/2209.05364
> (Mlodozeniec et al., 2024) Influence Functions for Scalable Data Attribution in Diffusion Models, https://arxiv.org/abs/2410.13850
> (Grosse et al., 2023) Studying Large Language Model Generalization with Influence Functions, https://arxiv.org/abs/2308.03296

---

### Note · Authors · 2025-08-12

We warmly thank the reviewers and AC for their engagement during a productive discussion period. This has witnessed two main improvements.

1. **Extra experiments**. Whilst the reviewers appreciate that the chief contributions of this paper are mathematical, we have added new larger scale experiments with ViTs and diffusion models to more clearly illustrate our points. These confirm the real-world relevance and applicability of distributional data attribution, and that our proposed algorithms can outperform their classical counterparts.
2. **Sharper messaging**. The paper covers a lot of ground and includes considerable theoretical depth. Some of our contributions and the justification for their importance was initially unclear to reviewers -- e.g. the distributional interpretation of ‘standard’ IFs, and the relationship between the two theorems. We resolved these by refactoring the introduction and communicating the claimed contributions more crisply -- see sTvu and wFZD. The reviewers appear to agree that the new presentation reads better.

Thank you for all the discussions and helping us strengthen the paper,
The authors.

---

### Decision · Program_Chairs · 2025-09-17

**Decision:**

Accept (spotlight)

**Comment:**

The paper introduces distributional training data attribution (d-TDA), a new framework for data attribution that explicitly accounts for the stochasticity in deep learning training. Traditional methods typically ignore the randomness from initialisation and data batching, providing a single point estimate of an example's influence. In contrast, d-TDA aims to predict how the entire distribution of model outputs changes when a training example is removed. The authors demonstrate that this approach can identify influential examples missed by conventional mean-based methods, such as those that primarily affect model variance. A key theoretical contribution is a new mathematical justification for influence functions (IFs). The paper shows that IFs emerge naturally from the d-TDA framework as a long-training-time limit of unrolled differentiation, without requiring the restrictive convexity assumptions of prior work.

Strengths. The paper's primary strength is its novel and well-motivated conceptual reframing of the data attribution problem. It addresses the fundamental issue of stochasticity in modern model training. The theoretical work is rigorous and provides a significant new justification for the effectiveness of influence functions.

Weaknesses. The initial submission's primary weaknesses were a lack of empirical validation on large-scale models and some ambiguity in its core message. Reviewers found it unclear whether the paper proposed a new practical algorithm to replace influence functions or a theoretical device to justify them. These issues were substantially addressed during the rebuttal period.

Discussion. The discussion period was smooth and highly constructive. Reviewers raised concerns about the limited scale of the experiments and the clarity of the paper's contributions. The authors added new experiments on ViTs and diffusion models. They also agreed to sharpen the paper's messages.

Why accept? New theoretical grounding for influence functions. The authors were responsive to feedback. Identifying and addressing a novel problem.

PS. "A Bayesian Approach To Analysing Training Data Attribution In Deep Learning" from NeurIPS 2023 is also highly relevant prior work -- recommend reading it.